# DNase II mediates a parthanatos-like developmental cell death pathway in *Drosophila* primordial germ cells

Lama Tarayrah-Ibraheim[1], Elital Chass Maurice[1], Guy Hadary[1], Sharon Ben-Hur[1], Alina Kolpakova[1], Tslil Braun[1], Yoav Peleg[2], Keren Yacobi-Sharon[1] & Eli Arama [1✉]

During *Drosophila* embryonic development, cell death eliminates 30% of the primordial germ cells (PGCs). Inhibiting apoptosis does not prevent PGC death, suggesting a divergence from the conventional apoptotic program. Here, we demonstrate that PGCs normally activate an intrinsic alternative cell death (ACD) pathway mediated by DNase II release from lysosomes, leading to nuclear translocation and subsequent DNA double-strand breaks (DSBs). DSBs activate the DNA damage-sensing enzyme, Poly(ADP-ribose) (PAR) polymerase-1 (PARP-1) and the ATR/Chk1 branch of the DNA damage response. PARP-1 and DNase II engage in a positive feedback amplification loop mediated by the release of PAR polymers from the nucleus and the nuclear accumulation of DNase II in an AIF- and CypA-dependent manner, ultimately resulting in PGC death. Given the anatomical and molecular similarities with an ACD pathway called parthanatos, these findings reveal a parthanatos-like cell death pathway active during *Drosophila* development.

[1] Department of Molecular Genetics, Weizmann Institute of Science, Rehovot 76100, Israel. [2] The Israel Structural Proteomics Center (ISPC), Weizmann Institute of Science, Rehovot 76100, Israel. ✉email: eli.arama@weizmann.ac.il

Programmed cell death (PCD) is a cell suicide process fundamental for the development and homeostasis of the organism, and malfunction of this process is associated with the pathogenesis of multiple diseases[1–5]. Apoptosis, the major form of PCD during animal development, is mediated by the activation of a unique family of cysteine proteases called caspases[6–8]. However, studies in the past several decades suggest that cells can sometimes activate caspase-independent alternative cell death (ACD) pathways[9,10]. About a dozen mechanistically distinct ACD pathways have been described in different experimental systems and studied mainly under non-physiological conditions[11]. ACDs have received considerable attention in recent years because of their involvement in the pathogenesis of a variety of diseases, leading to the perception that intervention in these pathways could be a promising avenue for developing new therapeutic strategies[12–14]. However, many questions regarding the in vivo biology of ACD pathways remain open, as it is inevitable that studies in cell culture cannot fully reproduce the complexity of the in vivo reality, leading, at times, to inaccurate notions[15]. On the other hand, although fascinating, only a few physiological ACD paradigms have been described in complex metazoan organisms, e.g. linker cell death in *C. elegans*[16,17], autophagic cell death in the *Drosophila* salivary glands[18] and midgut[19], germ cell death of *Drosophila* spermatogonia[20,21], and phagoptosis of the germline nurse cells in *Drosophila*[22]. Furthermore, the connections, if any, between these paradigms and the non-physiological ACDs are not always clear, and for most of the ACDs, an equivalent physiological system has not been identified[9–11].

Parthanatos is an ACD pathway that has been almost exclusively investigated under non-physiological conditions in mammalian cells, and is distinct from apoptosis, autophagy or necrosis at both the molecular and biochemical levels[23,24]. Parthanatos is triggered by overexpression (OE) of the DNA damage-sensing enzyme, Poly(ADP-ribose) polymerase-1 (PARP-1), or its activation following DNA damage caused by genotoxic stress or excitotoxicity[23–25]. Following PARP-1 activation, the mitochondrial protein, Apoptosis Inducing Factor (AIF), is released to the cytosol, where it associates with and facilitates nuclear translocation of the Deoxyribonuclease (DNase) Macrophage Migration Inhibitory Factor (MIF)[26–28]. Parthanatos has been implicated in the pathogenesis of several important diseases, including Parkinson's disease, stroke, heart attack, diabetes, and ischemia reperfusion injury in numerous tissues[29–31]. However, it remains unknown whether cell death by parthanatos is only limited to stress or pathological conditions, or if it might also operate during normal development and homeostasis of the organism.

Here, we report the characterization of a caspase-independent form of PCD in *Drosophila*, through which about 30% of the primordial germ cells (PGCs) are normally eliminated during early embryogenesis. Detailed analysis of the underlying mechanisms revealed striking resemblance to parthanatos; PGC death pathway is mediated by AIF-dependent nuclear translocation of the lysosomal nuclease, DNase II, and the consequent DNA damage induced PARP-1 activation. Given the rarity of developmental ACD pathways identified in metazoan model organisms, these findings may extend our understanding of the connections between the non-physiological and physiological ACD pathways, as well as when and why ACD may sometimes be advantageous over apoptosis.

## Results

**PGCs undergo caspase-independent cell death.** Primordial germ cells (PGCs) arise from germline progenitors, called pole cells, initially found at the posterior pole of the early *Drosophila* embryo[32]. During gastrulation, the PGCs are carried as a loose cluster into the posterior midgut pocket (embryonic stage [ES] 9) from where they migrate across the midgut epithelium near the embryo midline (ES 10; Fig. 1a). The PGCs disperse from this cluster, sort bilaterally (ES 11) and migrate toward somatic gonadal precursors (ES 12), eventually compacting into two round gonads (ES 13; Fig. 1a)[33,34]. However, not all PGCs specified at early embryogenesis successfully migrate from the midline position to the gonads, and several reports showed that these cells are eliminated by cell death[34–39]. Interestingly, previous attempts to block PGC death through inhibition of apoptosis, including genetic inactivation of the Inhibitor of apoptosis (IAP) protein antagonists (the *reaper* family genes) and overexpression (OE) of the baculovirus effector caspase inhibitor protein p35, and the *Drosophila* IAP proteins, Diap1 and Diap2, as well as OE of a dominant-negative (DN) form of the *Drosophila* caspase-9 homolog Dronc, have all failed in this regard, suggesting divergence from the conventional apoptotic program[35–37,39].

To explore the mechanisms underlying PGC death, we implemented a unified quantitative method which has been used by several groups to evaluate the levels of PGC death in different *Drosophila* strains and mutants[35–37,39]. The basis for this approach is that the number of PGCs is relatively small and can be readily visualized by staining with an anti-Vasa antibody, allowing for manual counting of their numbers before cell death induction, when all the PGCs are still dispersed at the embryo midline (ES 10), and after cell death of the aberrantly migrating PGCs is almost completed (ES 13; Fig. 1a, b). Since during these stages, the PGCs neither divide[40] nor transdifferentiate[41], the difference in the number of PGCs between ES 10 and 13 accurately reflects the number of dying PGCs. To define the average levels of PGC death, we examined embryos from three different standard *D. melanogaster* laboratory reference strains, *yellow¹ white¹¹¹⁸* (*yw*), Oregon R (OR), and Canton-S (CS). Whereas within each of these fly strains, the number of PGCs at ES 10 was relatively constant (n ± 3), it could vary considerably among embryos of the different strains at the same ES (Fig. 1c). In contrast, the relative decrease in the number of PGCs between ES 10 and 13 (~30%) was highly conserved among all of the examined fly strains, allowing for comparative measurements of PGC death levels among different genetic backgrounds (Fig. 1c). The dying midline PGCs can sometimes be also directly visualized at advanced demolition stages by virtue of their condensed and distorted morphology, as well as the reduction in the Vasa staining signal (arrows in Fig. 1b). Since by ES 13 only a few such dying PGCs could be detected, implying that most of the midline PGCs already undergo cell death at earlier stages, we examined the time window during which most of the midline PGCs undergo cell death by counting the PGC numbers at ES 10, 11, 12, and 13. Whereas condensed and distorted dying PGCs could be detected at all these stages, the majority underwent cell death between ES 10 and 12 with a small peak at ES 11 (Fig. 1d). These findings are also in agreement with a previous report showing that most of PGC death occurs between ES 10 and 11[39].

Collectively, we conclude that between ES 10 and 13, the PGCs are divided into two subsets, one of which migrates to the gonads (encompassing around two-thirds of the PGCs and are henceforth referred to as the gonadal subset), and the second of which remains near the midline and undergoes cell death in an asynchronous manner (encompassing about one-third of the total PGCs and are henceforth referred to as the midline subset).

Using this assay, we initially wanted to confirm that PGC death can indeed proceed when caspase activity is compromised. For this, we used the *nos-Gal4-VP16* driver[42] to overexpress potent inhibitors of the apoptotic caspase activity

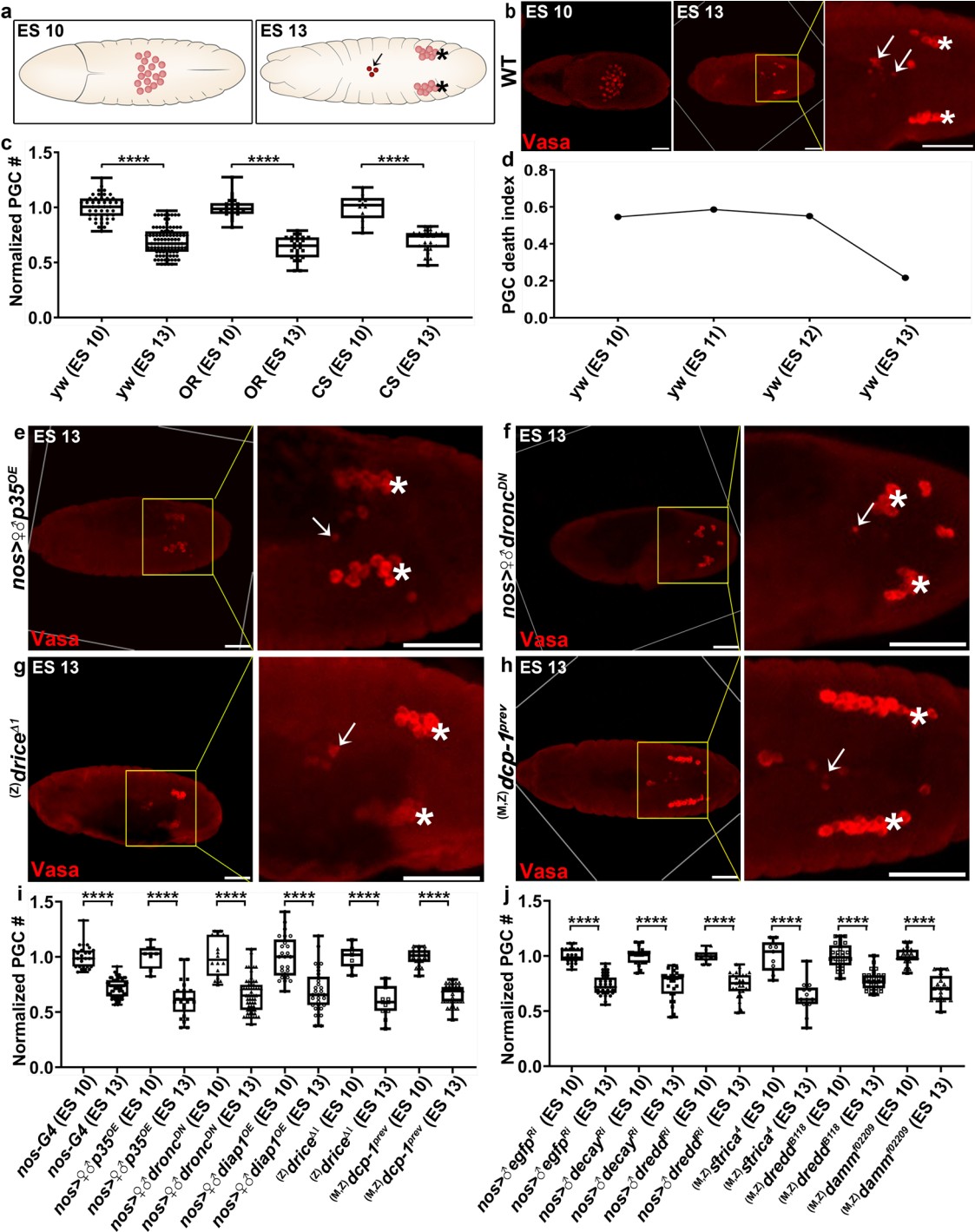

and/or specific RNA interference transgenes (Ri), as well as used genetic mutants when possible, in order to inactivate the seven apoptotic and non-apoptotic *Drosophila* caspases. As previously reported, OE of the potent caspase inhibitory proteins, p35, Diap1 and Dronc[DN], all failed to attenuate PGC death (Fig. 1e, f, i). Furthermore, PGC death proceeded normally when each of the seven *Drosophila* caspases, as well as the major apoptotic corpses engulfment receptor homolog of CED-1, Draper (*drpr[Δ5]*)[43], were inactivated, either following PGC-specific knockdown or in maternally and/or zygotically mutant embryos (Fig. 1g–j and Supplementary Fig. 1m). Altogether, these findings confirm that PGC death is a *bona fide* caspase-independent cell death pathway.

**DNase II mediates PGC death but not PGC migration.** To identify the pathway by which PGCs undergo cell death, we first tested for possible involvement of an ACD pathway called germ cell death (GCD), which operates in the adult *Drosophila* testis[20]. We inactivated four genes encoding major mediators of GCD: The mitochondrial serine protease HtrA2/Omi and Endonuclease G (EndoG), and the lysosomal endonuclease DNase II and protease Cathepsin D (CathD). Critically, whereas both maternal and zygotic inactivation of Omi, EndoG, and CathD had no significant effects on the levels of PGC death (Supplementary Fig. 1a–d, g), a complete block of PGC death was detected in embryos maternally, but not zygotically, mutant for a strong hypomorphic allele of DNase II (*dnaseII[lo]*), as well as upon

**Fig. 1 PGCs die through a caspase-independent pathway. a** Dorsal view illustrations of *Drosophila* embryos at ES 10 (left) and ES 13 (right). During these stages, PGCs (red) migrate from the midline region (ES 10) to the gonadal sites (asterisks; ES 13). About 30% of the PGCs fail to migrate and undergo cell death (arrow). **b**, **e–h** Representative images of embryos of the indicated genotypes and embryonic stages stained to visualize the PGCs (Vasa; red). The outlined areas (yellow squares) are magnified in the right panels, presenting the midline region with the dying PGCs (arrows pointing at highly condensed dying PGCs with reduced Vasa signal; asterisks indicate gonadal PGCs). Note that since the demolition process is not fully synchronous, some highly condensed dying PGCs can still be visualized in the midline region of ES 13 embryos. Scale bars 50 µm. **c**, **i**, **j** Quantification of PGC death levels in embryos of the indicated genotypes by normalizing the total PGC number in each individual embryo at ES 10 and ES 13 to the average PGC number at ES 10 (the highly condensed dying PGCs detected at ES 13 are considered dead cells and thus omitted from our calculations). All data points, including outliers, were presented in box plot format where the minimum is the lowest data point represented by the lower whisker bound, the maximum is the highest data point represented by the upper whisker bound, and the center is the median. The lower box bound is the median of the lower half of the dataset while the upper box bound is the median of the upper half of the dataset. Each dot corresponds to the number of PGCs in a single embryo to reflect *n* number, where *n* = number of examined biologically independent embryos. ****$p < 0.0001$, Student's *t*-test, one-sided distribution. **d** PGC death index reflects the frequency of detection of morphologically condensed/distorted dying PGCs at the relevant embryonic stage. **b**, **c** Regardless of the variations in the number of PGCs at ES 10, which depends on the fly strain background (shown are 3 different WT strains, yw, OR, and CS, containing, at ES 10, 27 ± 3, 34 ± 4, and 33 ± 3 PGCs on average, respectively), the relative percentage of the aberrantly migrating midline PGCs undergoing cell death remains constant (i.e. ~30%). **e–j** PGC death proceeds in the absence of the apoptotic and non-apoptotic caspases. ♀, female; ♂, male; (M) and (Z), maternal and zygotic mutant embryos, respectively.

PGC-specific knockdown of *dnaseII* (*dnaseII^Ri*; Fig. 2a–d). Importantly, inactivation of DNase II had no effect on normal migration and incorporation to the gonads of the gonadal subset (Fig. 2a, c, e). Furthermore, during subsequent embryonic stages, the ectopically surviving midline subset were scattered randomly in the posterior half of the mutant embryos, and could be readily detected even at ES 17 when the larval structures are already formed (Fig. 2f–j).

Taken together, these observations suggest that maternal *dnaseII* is cell autonomously required for PGC death, and that in its absence, the midline subset of PGCs survive throughout embryogenesis.

Previous studies have coupled between PGC migration across the midgut at ES 9/10 and the competence to undergo cell death. In particular, it was shown that in embryos mutant for regulators of PGC migration, the PGCs remained inside the gut and were resistant to cell death[44]. To explore for possible effects of DNase II inactivation on the ability of the midline subset of PGCs to transverse the midgut, we co-stained *dnaseII* mutant embryos with anti-Hb9 antibodies, which label the posterior midgut primordium[45], and with anti-Vasa antibodies to reveal the PGCs. Whereas PGCs that cross the midgut epithelium could be detected already at ES 10, it was still difficult to differentiate between the different subsets of PGCs at that stage (Supplementary Fig. 2a). However, starting at ES 11 and onward, the gonadal subset of PGCs, which already started sorting into two groups, and the surviving midline subset, became clearly distinct, and both were positioned outside of the midgut (Supplementary Fig. 2b–d). Therefore, migration of the midline PGC subset across the midgut is unaffected in the *dnaseII* mutants.

**DNase II is involved in PGC death upon Wunens inactivation.** Once PGCs transverse the midgut, they become dependent on survival signals provided maternally by the lipid phosphate phosphatases (LPPs), Wunen (Wun), and Wun2, which act redundantly in the germ cells and the soma to regulate PGC migration and cell death[35,36,44]. Interestingly, it has been shown that in embryos maternally mutant for *wun2* alone or double mutant for *wun2* and *wun*, PGCs undergo precocious non-apoptotic cell death, eliminating about 50% of the PGCs in the *wun2* mutants and almost all the PGCs in the double mutants[35,36,44]. We, therefore, asked whether DNase II might also mediate PGC death induced by the lack of maternal *wunens*. Analyzing a mutant allele of *wun2*, which was suggested to also act as a dominant-negative allele of *wun*, dubbed *wun2^N14* (henceforth referred to as *wunens^−/−*)[35,44], revealed that 40% of the maternally mutant *wunens^−/−* embryos displayed no PGCs already at ES 9/10, suggesting that they might

have died or transdifferentiated early in development (Fig. 3a, b, g). We then compared PGC numbers in ES 13 and 14 embryos laid by *wunens^−/−* deficient mothers and by *wunens^−/−* and *dnaseII^lo* double mutant mothers. Significantly, whereas half of the *wunens^−/−* embryos contained 2 PGCs on average and the other half had zero PGCs, 70% of the double mutant embryos contained 6 PGCs on average while 30% had zero PGCs (Fig. 3c–g). Since already at ES 9/10, 40% of the *wunens^−/−* embryos had no PGCs, these findings suggest that essentially all the double mutant embryos which started with regular numbers of PGCs at ES 9/10 contained surviving PGCs. Furthermore, the Vasa staining signal was much more intense in the double mutant surviving PGCs as compared with the *wunens^−/−* PGCs, further implying that the latter might be dying or transdifferentiating (Fig. 3c–f). Of interest, almost all of the surviving PGCs in the double mutants migrated to the gonads, further demonstrating that DNase II is not involved in PGC migration (Fig. 3d, f). Moreover, in a few double mutant embryos, the average number of surviving PGCs which migrated to the gonads (34 PGCs on average) was almost identical to the number of PGCs at ES 9/10 (33 PGCs on average), suggesting that in the absence of the *wunens*, both the gonadal and midline subsets of surviving PGCs have similar capacity to migrate to the gonads (Fig. 3d, f, g). Finally, not all PGCs survived in the double mutants, which may be attributed to a potent cell death signal triggered in the absence of survival signal from Wunens, and to the fact that the *dnaseII^lo* mutant is a hypomorph.

Overall, these observations suggest that DNase II is involved in PGC death induced by *wunens* deficiency, revealing a direct link between the developmental cell death pathway and the survival signaling pathway of PGC death.

**PGC death requires lysosomal leakage and is independent of macroautophagy.** To further explore the role of the lysosomal pathway in PGC death, we inactivated several lysosomal biogenesis proteins and catabolic proteases, and examined their effects on PGC death. PGC-specific knockdowns of the lysosomal cysteine proteases CathB and CathL, both significantly decreased PGC death levels (Supplementary Fig. 1d–g). Furthermore, PGC-specific knockdowns of the genetically linked lysosomal biogenesis proteins, the Vps18p homolog Deep-orange (Dor) and the Vps33A homolog Carnation (Car), completely blocked PGC death (Fig. 2k and Supplementary Fig. 1h, i). These observations imply that functional lysosomes mediate PGC death.

Two major mechanisms for the involvement of lysosomes in ACD-associated cellular degradation have been proposed: Sequestration of cellular contents to lysosomes by autophagy, or

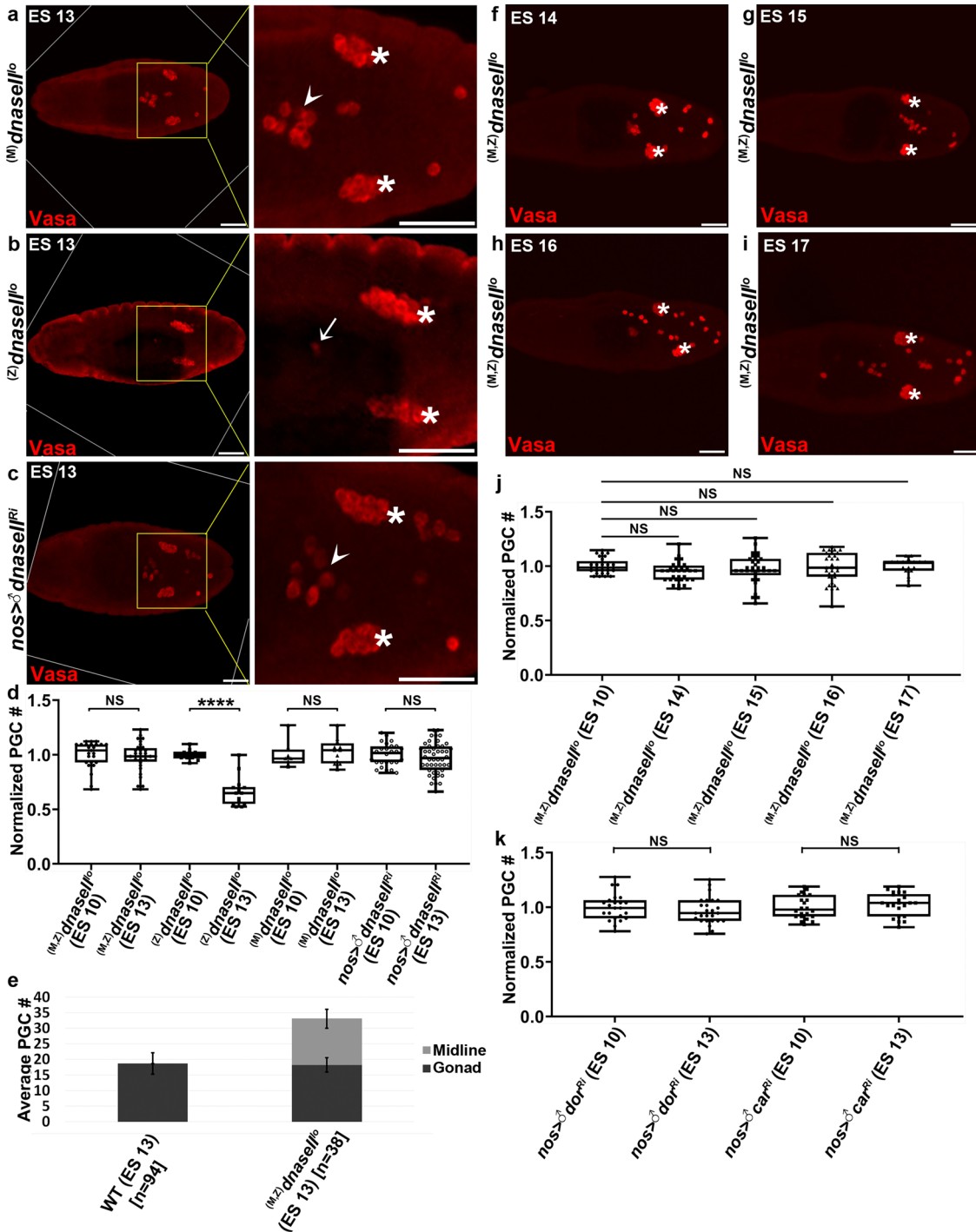

spillage of lysosomal catabolic enzymes to the cytosol, following the induction of lysosomal membrane permeabilization (LMP)[46]. We first tested for possible involvement of the autophagic pathway in PGC death. However, no significant effect on PGC death levels was detected when autophagy was genetically compromised, including in maternally mutant embryos trans-heterozygous for two null atg7 alleles (atg7[d77] and atg7[d14]), upon PGC-specific ectopic expression of a dominant-negative form of Atg1 (atg1[KQ13A]), or following RNAi-mediated knockdowns of atg7, atg5, atg6 and atg8a/LC3 (Supplementary Fig. 1j–l). In contrast, genetic inhibition of LMP, achieved by interference with several pathways previously reported to be involved in this process (Supplementary Fig. 3a; also summarized in ref. [47]), resulted in pronounced attenuation

of PGC death (Supplementary Fig. 3b–f). Taken together, we conclude that the lysosomes are major mediators of PGC death, presumably functioning through the release of their contents.

*Drosophila p53* mutants were previously shown to be defective in developmental PGC death, an effect that was attributed to both maternal and zygotic expressions of *p53*, but the underlying mechanisms remained elusive[39]. In light of previous reports implicating p53 as a positive regulator of LMP in mammalian cells[47], we monitored PGC death in ES 10 and 13 embryos both maternal and zygotic homozygous mutants for the *p53* null allele, *p53[5A-1-4]*, and in *p53* knockdown embryos. Whereas similar to the previous report, inactivation of p53 in the mutant and the knockdown embryos attenuated PGC death[39], this effect was only detected in 57% of the embryos (Fig. 3h–j). Interestingly, some of

**Fig. 2 Inactivation of the lysosomal endonuclease and compromised lysosomal biogenesis block PGC death. a–d** Maternally deposited *dnaseII* is required cell autonomously for PGC death. ES 13 embryos maternally (**a**), but not zygotically (**b**), mutant for a strong hypomorphic *dnaseII* allele, as well as embryos with PGC-specific *dnaseII* knockdown (**c**), display complete block in PGC death. Shown are representative images of embryos of the indicated genotypes stained and presented as in Fig. 1b. PGCs (Vasa; red). Note the intact morphology of the ectopically surviving midline PGCs in the maternal and knockdown mutants (arrowheads), as opposed to the highly condensed dying PGCs in the zygotic mutant embryo (arrow). Asterisks indicate gonadal PGCs. Scale bars, 50 μm. **d** Quantifications of PGC death levels in embryos corresponding to the genotypes in (**a–c**). All data points, including outliers, were presented in box plot format where the minimum is the lowest data point represented by the lower whisker bound, the maximum is the highest data point represented by the upper whisker bound, and the center is the median. The lower box bound is the median of the lower half of the dataset while the upper box bound is the median of the upper half of the dataset. Each dot corresponds to the number of PGCs in a single embryo to reflect *n* number, where *n* = number of examined biologically independent embryos. ****$p < 0.0001$; NS, non-significant; Student's *t*-test, one-sided distribution. **e** Ectopically surviving PGCs in *dnaseII* mutant embryos remain in the midline and do not migrate to the gonads. Quantification of the average number of PGCs that are present in the gonads versus the midline of WT and *dnaseII* mutant embryos at ES 13. *n* = number of examined biologically independent embryos. Data are presented as mean value±SD. **f–j** The ectopically surviving midline PGCs in the *dnaseII* mutant embryos persist throughout embryogenesis. PGCs (Vasa; red). **f–i** the ectopically surviving midline PGCs do not associate with the gonads, but rather remain scattered in the posterior half of the embryo. Scale bars, 50 μm. **j** Quantifications of PGC death levels in embryos corresponding to the genotypes in (**f–i**), calculated and presented as in (**d**). NS, non-significant; Student's *t*-test, one-sided distribution. Note that there is no reduction in PGC numbers in the *dnaseII* mutants as embryogenesis progresses. **k** PGC-specific knockdowns of the genetically linked lysosomal biogenesis genes *dor* and *car* block PGC death. Quantification of PGC death levels in embryos of the indicated genotypes calculated and presented as in (**d**). NS, non-significant; Student's *t*-test, one-sided distribution. Representative images are found in Supplementary Fig. 1h, i.

---

the *p53* mutant embryos displayed more PGCs at ES 13 (a maximum of 36 PGCs) as compared with the same mutants at ES 10 (a maximum of 29 PGCs), implying that some of the PGCs may undergo cell division between ES 10 and 13 in these mutants. We therefore conclude that p53 may be involved in several aspects of PGC biology, including PGC death.

**DNase II nuclear translocation requires functional genetic LMP pathways.** Our observations suggest a cell autonomous function for DNase II during PGC death, which implies translocation of this lysosomal nuclease to the nucleus. To test this hypothesis, we generated specific polyclonal antibodies against full-length *Drosophila* DNase II. The capacity of these antibodies to detect DNase II in situ was confirmed by staining WT and *dnaseII* mutant embryos, showing that the specific DNase II signal in WT PGCs was abolished in the mutant (Fig. 4a, b). In WT embryos, DNase II was highly expressed in all the PGCs at ES 10, such that these cells could be clearly distinguished from the other cells of the embryos by virtue of the strong DNase II signal (Fig. 4a). Closer examination of stained ES 10 PGCs revealed that in the majority of the PGCs DNase II was confined to small cytoplasmic vacuoles (Supplementary Fig. 4a, b), which we confirmed to be the lysosomes by co-staining for the lysosomal small GTPase membrane protein Arl8[48] (Fig. 4c). Importantly, whereas DNase II that is exiting the lysosomes could be already detected in some of the midline PGCs with regular morphology (revealed by partial mislocalization with Arl8; Fig. 4d), dying PGCs, revealed by their condensed and distorted morphology, displayed strong and almost complete nuclear DNase II localization, implying that translocation of DNase II to the nucleus may constitute one of the earliest events of PGC death (Fig. 4e, f, Supplementary Movie 1, and Supplementary Fig. 4a, c).

In order to translocate to the nucleus, DNase II must be released from the lysosomes. We therefore examined whether DNase II translocation to the nucleus might be affected in the LMP deficient mutants. Staining these mutants, as well as the *p53* mutant and knockdown embryos for DNase II revealed that all the ectopically surviving midline PGCs in these mutants displayed non-nuclear DNase II at ES 13 (Fig. 3k, l and Supplementary Fig. 3g–i). In contrast, 70% of the dying midline PGCs in WT embryos at ES 11 displayed nuclear DNase II (Supplementary Fig. 3i). Note that we compared between embryos of different developmental stages (WT, ES 11; mutants ES 13), as most of the dying PGCs in WT embryos are detected at ES 11 (Fig. 1d), while in the mutants, the ectopically surviving midline PGCs accumulate by ES 13. Overall, these findings demonstrate that DNase II nuclear translocation is dependent upon functional genetic LMP pathways.

**PGC death is associated with DNase II-dependent DNA breaks.** Optimal function of the lysosomal hydrolases usually requires the low pH environment of the lysosomes. We therefore asked whether DNase II might still be functional in the nuclei of dying PGCs. For this, we examined whether PGC death is associated with DNA fragmentation, and if so, whether this might be dependent on DNase II. Examining nuclei of dying PGCs stained with the DNA fluorescent dye Hoechst, revealed gradual loss of the fluorescent signal, concomitant with the progression in cellular demolition and the prominence of nuclear DNase II, suggesting that the DNA in the dying PGCs is highly fragmented (Fig. 5a and Supplementary Movie 1). Furthermore, at around early-to-mid cell demolition stages, the dying PGCs stained positively with specific antibodies against phosphorylated histone H2A variant (γ-H2Av; the *Drosophila* equivalent of the mammalian γ-H2AX), an early response modification to the presence of DNA double-strand breaks (DSBs) (Fig. 5b, d)[49]. Importantly, no such staining was detected in ectopically surviving PGCs at the midline of *dnaseII*[lo] mutant embryos, implying that self-inflicted DNA double-strand breaks are mediated by DNase II during PGC death (Fig. 5c, d). It is interesting to note that although detected at a very low frequency, a few gonadal PGCs that are positive to γ-H2Av are also observed, which is consistent with a report observing 1-2 PGCs (on average) undergoing cell death in the gonads[37].

A classical assay for direct in situ detection of DNA fragmentation during apoptosis is the terminal deoxynucleotidyl transferase-mediated dUTP nick-end labeling (TUNEL). However, consistent with previous reports indicating that cells dying in response to Wunens-mediated signals were negative for TUNEL staining[35,37], the developmentally dying PGCs were completely negative for TUNEL during the entire death process, although they displayed pronounced TUNEL labeling when induced to undergo precocious apoptosis by OE of the proapoptotic *reaper* family gene *hid* (Supplementary Fig. 5a, b). These observations suggest that PGC death is normally associated with DNA cuts distinct from those generated during apoptosis.

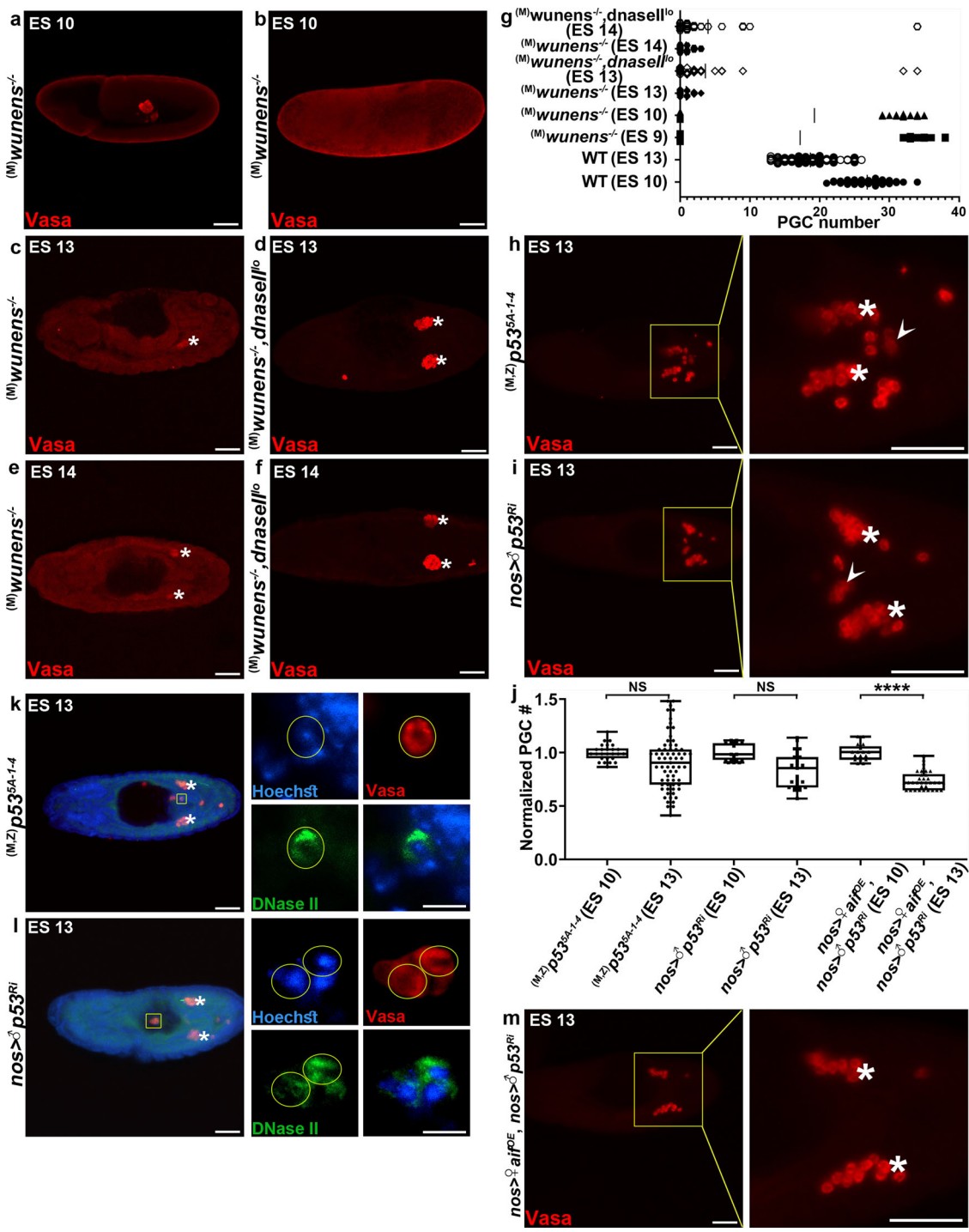

Whereas the TUNEL assay is designed to label DNA fragments with exposed 3' hydroxyl (OH) groups generated by type I DNases, labeling of DNA fragments with exposed 5'-OH and 3'-PO4, which are produced by type II DNases (such as DNase II and Topoisomerase I [Top I]), requires a distinct assay based on Top I-mediated ligation of a fluorescently labeled oligonucleotide[50]. Indeed, applying this assay to WT embryos revealed specific nuclear labeling in some of the dying midline PGCs, and not in the gonadal PGCs (Fig. 5e). Moreover, in the *dnaseII*[lo] mutant embryos, no labeling was detected in the ectopically surviving midline PGCs, consistent with direct involvement of DNase II in eliciting this DNA fragmentation (Fig. 5f). Consistently, no Top I-associated

labeling was detected in PGCs that were induced to undergo precocious apoptosis, confirming the specificity of this assay for DNase II type cuts (Fig. 5g).

## DNA damage and activation of the DDR pathway mediate PGC death.

As opposed to apoptosis, in which DNA fragmentation marks the end-stage and ultimate demise of the cell, our findings suggest that the DNase II mediated self-inflicted DNA double-strand breaks is an early step in triggering PGC death. To independently test this idea, we sought to induce mild DNA damage in the PGCs by taking advantage of the *Chlamydomonas reinhardtii* homing endonuclease I-*Cre*I, which was shown to induce multiple DNA breaks at a defined location on the *Drosophila* sex

**Fig. 3 The interplay between DNase II and Wunens in PGC survival, and DNase II and p53 in PGC death. a–g** The *dnaseII*[lo] mutant partially attenuates PGC death induced by the lack of maternal *wunens*. Shown are representative images of embryos maternally lacking the *wunens* (laid by *wun2*[N14]/*Df* mutant mothers) at ES 10 (**a**, **b**), ES 13 (**c**) and ES 14 (**e**), as well as embryos maternally double mutant for both *wunens* and *dnaseII* at ES 13 (**d**) and ES 14 (**f**), stained to visualize the PGCs (Vasa; red). Asterisks indicate gonadal PGCs. Scale bars, 50 µm. The corresponding quantifications of total PGC numbers as compared to WT embryos are shown in (**g**). Mean PGC number is shown as a line in the middle of each data set, each dot is a single embryo to reflect *n* number, where *n* = number of examined biologically independent embryos (WT ES10 [*n* = 46]; WT ES13 [*n* = 94]; (M)*wunens*−/− ES9 [*n* = 34]; (M)*wunens*−/− ES10 [*n* = 35]; (M)*wunens*−/− ES13 [*n* = 30]; (M)*wunens*−/−, *dnaseII*[lo] ES13 [*n* = 34]; (M)*wunens*−/− ES14 [*n* = 24]; (M)*wunens*−/−, *dnaseII*[lo] ES14 [*n* = 36]). See detailed description of the results in the main text. **h–j** p53 is cell autonomously required for PGC death. Representative images of a *p53* null mutant embryo (**h**) and an embryo with PGC-specific *p53* knockdown (**i**), both at ES 13, stained and presented as in Fig. 1b. PGCs (Vasa; red). Asterisks indicate gonadal PGCs. Arrowheads point at ectopically surviving PGCs. Scale bars, 50 µm. **j** The corresponding quantifications of PGC death levels. All data points, including outliers, were presented in box plot format where the minimum is the lowest data point represented by the lower whisker bound, the maximum is the highest data point represented by the upper whisker bound, and the center is the median. The lower box bound is the median of the lower half of the dataset while the upper box bound is the median of the upper half of the dataset. Each dot corresponds to the number of PGCs in a single embryo to reflect *n* number, where *n* = number of examined biologically independent embryos. ****$p < 0.0001$; NS, non-significant; Student's *t*-test, one-sided distribution. Note that some of the p53 mutant embryos contained more PGCs at ES 13 than in ES 10, implying an involvement of p53 in controlling cell division at these stages. **k**, **l** Nuclear translocation of DNase II is blocked in ectopically surviving *p53* mutant (**k**) and knockdown (**l**) PGCs. Shown are ES 13 embryos stained to visualize the PGCs (Vasa; red), DNase II (green), and DNA (Hoechst, blue). The outlined areas (yellow squares) are magnified in the corresponding panels on the right. Asterisks indicate gonadal PGCs. Scale bars, 50 µm. Surviving midline PGCs are outlined (yellow circles). Note the non-nuclear DNase II localization in the surviving midline PGCs. Scale bars, 10 µm. The corresponding quantifications are presented in Supplementary Fig. 3i. Note that images of WT embryos stained with the anti-DNase II antibodies are shown in Fig. 4. **m** *aif* OE restores PGC death levels in *p53* mutants. Shown is a representative image of a *p53* mutant embryo at ES 13 with PGC-specific *aif* OE stained, presented, and annotated as in (**h**, **i**). Asterisks indicate gonadal PGCs. Scale bars, 50 µm. The corresponding quantification is presented in (**j**). Note that OE of *aif* in an otherwise WT background does not affect PGC death levels; Fig. 7b).

chromosomes[51]. ES 8 or 10 embryos carrying the heat-inducible *70I-CreI* transgene were treated with or without heat shock (1 hr, 37°), matured at 25° to ES 10 or 13 (respectively), and stained to visualize the PGCs. Whereas even without treatment, the embryos carrying the transgene displayed precocious PGC death at both ES 10 and 13 as compared with corresponding WT embryos (presumably due to leakiness of the transgene minimal promoter), this trend became highly significant after the heat shock treatment (Supplementary Fig. 5c–g). Of particular relevance, as opposed to the *hid*-overexpressing apoptotic PGCs that were readily labeled by TUNEL and activated caspases (cleaved Dcp-1), the dying I-*CreI* overexpressing PGCs were negative for both TUNEL and activated caspases (Supplementary Fig. 5h–k). Therefore, deliberate induction of mild DNA damage can trigger non-apoptotic cell death in PGCs.

Our observations suggest that DNase II-induced DNA fragmentation is a key event in the pathway leading to PGC death. Phosphorylation of the histone variant H2Av, which we observed in dying PGCs, is an early step in activation of the DNA damage response (DDR) following DNA breaks[49], raising the possibility that the DDR pathway is involved in PGC death. To address this notion, we knocked down major components in the two branches of the DDR pathway, ATM/Chk2 and ATR/Chk1 (see also the scheme in Fig. 6a)[52]. Significantly, PGC-specific knockdown of several components in the ATR/Chk1 branch, including the ATM-related kinase ATR, the ATR activator TopBP1, and the major ATR effector in the DNA damage checkpoint, Chk1, all attenuated PGC death, while knockdown of ATM/Chk2 branch components had no effect (Fig. 6b–g). Note that despite exhibiting about 40% penetrance of PGC death attenuation in the *chk1* knockdown embryos, the quantification difference between the PGC numbers at ES 10 and 13 still appears as significant (red asterisks). This is attributed, at least in part, to the fact that at ES 10, some of these embryos displayed exceptionally high numbers of PGCs (up to 39 PGCs), suggesting a role of the DDR in earlier stages of PGC specification and/or division. Finally, inactivation of p53, which is also an effector of the ATM/Chk2 pathway during irradiation induced apoptosis, attenuates PGC death (Fig. 3j and[39]), although, in this context, the effect is attributed to the function of p53 in promoting LMP[47].

Recent reports illuminate important crosstalk between the DDR and other checkpoint mechanisms critical for preventing genome instability, such as the spindle assembly checkpoint (SAC) and the DNA replication checkpoint[53,54]. Consistent with this idea, OE of two key components in these pathways, Orc2 (a component of the origin recognition complex [ORC]) and Bub3 (a main component of the SAC), both significantly attenuated PGC death (Supplementary Fig. 6a–c). Moreover, PGC-specific *orc2* knockdown which did not affect PGC death levels, was sufficient to restore cell death of the ectopically surviving PGCs in the *dnaseII*[lo] mutant embryos, implying that activation of the DDR is sufficient to promote PGC death, even in the absence of DNA damage (Supplementary Fig. 6d).

Our genetic data supports a sequence of events, in which relocated nuclear DNase II elicits DNA damage, which in turn leads to DDR activation and cell death. To further test this model, we compromised the DDR in early embryos and monitored the localization of DNase II in the ectopically surviving PGCs. DNase II was detected in the nuclei of ectopically surviving PGCs, but not the gonadal PGCs, following either knockdown of *chk1* or OE of Bub3, indicating that DDR activation is indeed downstream of DNase II nuclear translocation (Supplementary Fig. 6e–j). Of interest, the accumulation of DNase II in the nuclei of ectopically surviving PGCs was more pronounced following OE of Bub3 than knockdown of *chk1*, suggesting that Bub3 interferes with the DDR downstream of *chk1*, and that activation of the ATR/Chk1 branch of the DDR is required for further translocation of DNase II to the nucleus in a positive feedback loop (Supplementary Fig. 6g, j). Consistent with the latter idea, whereas WT dying midline PGCs at mid-to-late demolition stages readily displayed both nuclear DNase II and γH2Av staining, these were much rarer in the *chk1* knockdown embryos (1 out of 50), displaying only partial nuclear localization of DNase II (Supplementary Fig. 4d, e).

**AIF cooperates with DNase II to promote PGC death.** It has been previously suggested that the mitochondrial protein AIF, which does not exert nuclease activity on its own, can none-theless promote chromatin condensation and internucleosomal DNA fragmentation in vitro, a process sometimes referred to as chromatinolysis[55]. Furthermore, AIF has been implicated in

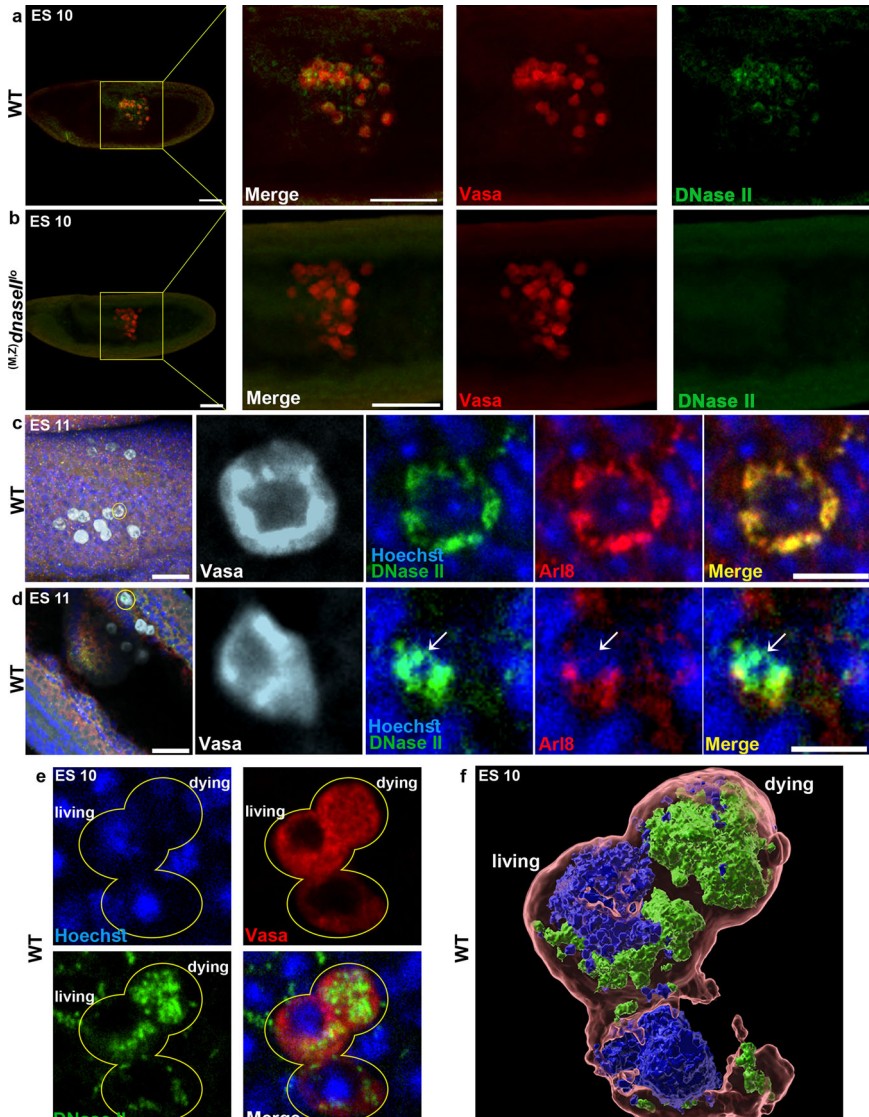

**Fig. 4 DNase II is released from the lysosomes and translocates to the nucleus in dying PGCs. a,b** Representative images of ES 10 WT (**a**) and *dnaseII* mutant (**b**) embryos stained to visualize the PGCs (Vasa; red) and DNase II (green). Note the lack of DNase II signal in the PGCs in the *dnaseII* mutant, as opposed to the strong signal in the WT counterparts. The outlined area (yellow square) is magnified in the right panels. Scale bars, 50 μm.
**c**, **d** Representative images of ES 11 WT embryos stained to visualize the PGCs (Vasa, white), DNase II (green), lysosomes (Arl8, red), and DNA (Hoechst, blue). Scale bars, 30 μm. The outlined cells (yellow circles) are magnified in the right panels. Scale bars, 10 μm. Note that in living PGCs, DNase II is unequivocally localized to the lysosomes (**c**), whereas at early cell death stages, DNase II is released from the lysosomes (**d**). **e**, **f** DNase II translocation to the nucleus in the dying PGCs is associated with DNA fragmentation. Living and dying PGCs in an ES 10 WT embryo (**e**) and a 3D rendering of this image (**f**) stained to visualize the PGCs (Vasa, red), DNase II (green), and the DNA (Hoechst, blue). Note the nuclear localization of DNase II and the associated dramatic reduction in the intensity of the Hoechst staining (typical of highly fragmented DNA) in the dying PGC. See also the accompanying Supplementary Movie 1 for full 3D visualization of the image presented in (**f**). Scale bars in **e** 2 μm; **f** 1 μm.

some forms of caspase independent ACD pathways through associating with and facilitating the nuclear translocation and enhanced activity of different nucleases, including EndoG in *C. elegans* and LEI/LDNaseII and MIF in mammalian cells[27,28,55–58]. Given the critical involvement of DNase II in eliciting DNA fragmentation and PGC death, we set up to explore whether AIF might also function to facilitate DNase II nuclear translocation and PGC death. Since homozygous mutant flies for a null allele of *aif* (*aif^T52*) are embryonic lethal, we examined both *aif* hypomorphic mutants, carrying the null allele in trans to a weaker *aif* allele (*aif^T2*), and PGC-specific *aif* knockdown embryos. Importantly, both the *aif^T52/T2* maternally mutant embryos and the *aif* knockdown embryos displayed significant attenuation in PGC death, indicating that similar to

DNase II, maternally supplied *aif* is cell autonomously required for PGC death (Fig. 7a, b).

We next explored possible genetic interactions between *aif* and *dnase II*. Since the *dnaseII^lo* allele encodes a weakly functional variant of DNase II[59], we first asked whether OE of AIF could compensate for the reduced DNase II activity in this genetic background. Notably, PGC-specific full-length AIF OE, which by itself did not increase PGC death levels in WT embryos, restored PGC death to almost normal levels in *dnaseII^lo* homozygous mutant embryos, as well as in *aif* knockdown embryos (which served as control), implying that AIF indeed cooperates with DNase II in mediating PGC death, and that PGC death is highly sensitive to the levels of both AIF and DNase II (Fig. 7b–d). Likewise, AIF OE restored PGC death to almost normal levels in

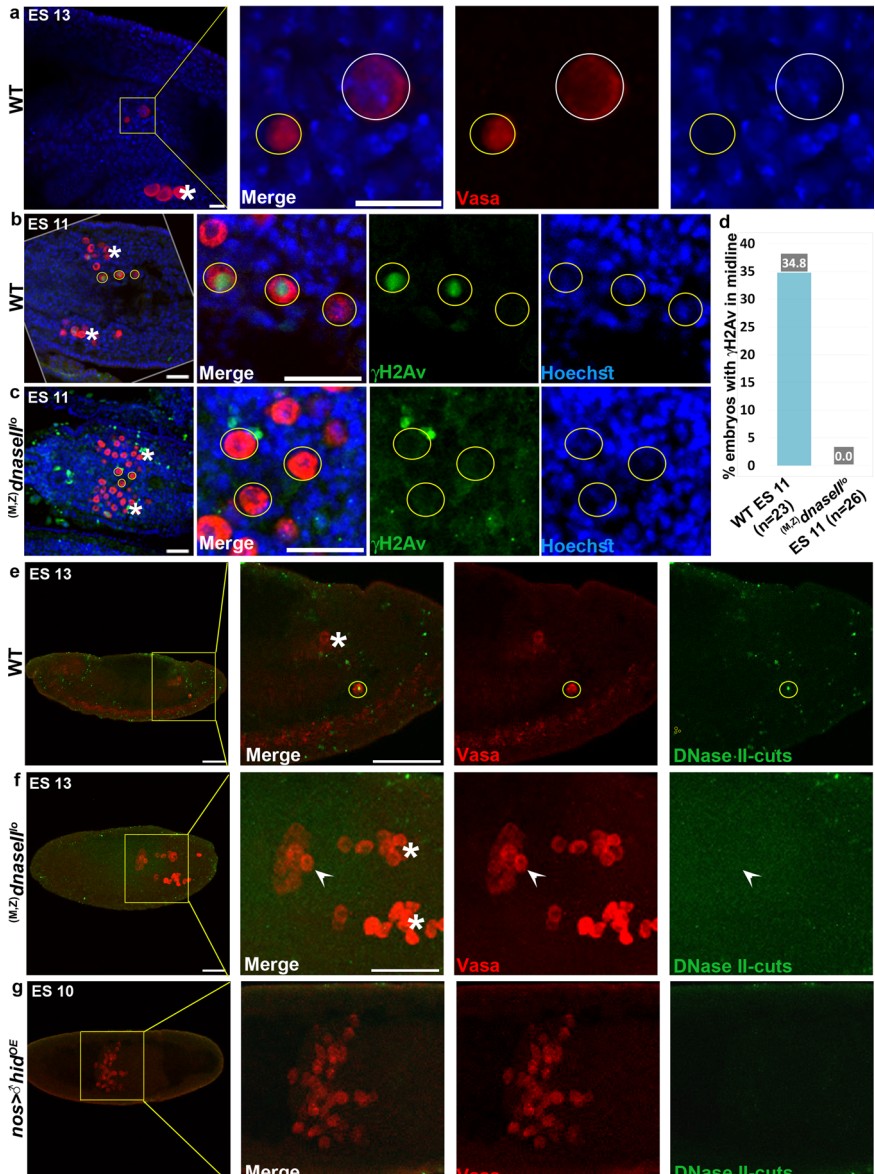

**Fig. 5 PGC death is associated with DNase II-dependent DNA breaks. a** DNA of dying PGCs becomes highly fragmented. A WT embryo at ES 13 stained to visualize the PGCs (Vasa; red) and the DNA (Hoechst; blue). The area of the midline PGCs is magnified in the right panels. Shown, are two dying midline PGCs during early (white circle) and late (yellow circle) cell death stages. Hoechst is readily detected during the early PGC death stage, and its signal fades away at advanced PGC death stages, typical of extensive DNA fragmentation. The asterisk indicates gonadal PGCs. Scale bars, 10 µm. **b**-**d** Dying PGCs exhibit DNase II-dependent DNA breaks. Images of the midline areas of WT (**b**) and *dnaseII* mutant (**c**) embryos, at ES 11, stained to visualize phosphorylation of H2Av (γH2Av, green; an early event following DSBs), PGCs (Vasa; red), and DNA (Hoechst; blue). Circled are three dying midline PGCs at different cell death stages displaying varying levels of γH2Av (**b**). In contrast, ectopically surviving midline PGCs in *dnaseII* mutants displayed no γH2Av signal (circled in **c**). The asterisks indicate gonadal PGCs. Scale bars, 20 µm. **d** Corresponding quantifications of the percentage of embryos displaying γH2Av staining. The percent value is indicated above each column. *n*, number of examined embryos. **e**–**g** Developmentally dying PGCs, but not ectopically surviving *dnaseII* mutant PGCs or dying apoptotic PGCs, exhibit DNase II-type DNA breaks. Representative images of WT (**e**) and *dnaseII* mutant (**f**) embryos, at ES 13, as well as an ES 10 embryo with PGC-specific OE of *hid* (**g**), all labeled by a Top I-mediated ligation assay to visualize DNase II-type DNA cuts (green). PGCs (Vasa; red). The outlined areas (yellow squares) are magnified in the right panels. Circled, is a dying PGC positively labeled for DNase II-type DNA cuts. An arrowhead is pointing at unlabeled ectopically surviving PGCs. Asterisks indicate gonadal PGCs. Note that Hid-induced apoptotic PGCs readily stain for TUNEL and cleaved caspase (Supplementary Fig. 5b, h, j). Scale bars, 50 µm.

*p53* knockdown embryos, linking between other components in the PGC death pathway and AIF (Fig. 3j, m)

The cooperation between AIF and DNase II was further examined by testing their ability to trigger non-apoptotic precocious PGC death. Taking advantage of the findings that, similar to AIF OE, PGC-specific DNase II OE could not trigger precocious PGC death in WT embryos on its own (although it could restore normal PGC death levels in *dnaseII^{lo}* mutant

embryos [Supplementary Fig. 7a]), we tested the effect of double OE on PGC death. PGC-specific OE of both DNase II and AIF triggered precocious PGC death already at ES 10, such that some of the ES 13 embryos were almost completely devoid of PGCs (Supplementary Fig. 7b–d). It is noteworthy that despite the dramatic reduction in the pool of PGCs at ES 10, still only two-thirds of the remaining PGCs migrated towards the gonadal somatic precursor cells, indicating that induction of cell death by

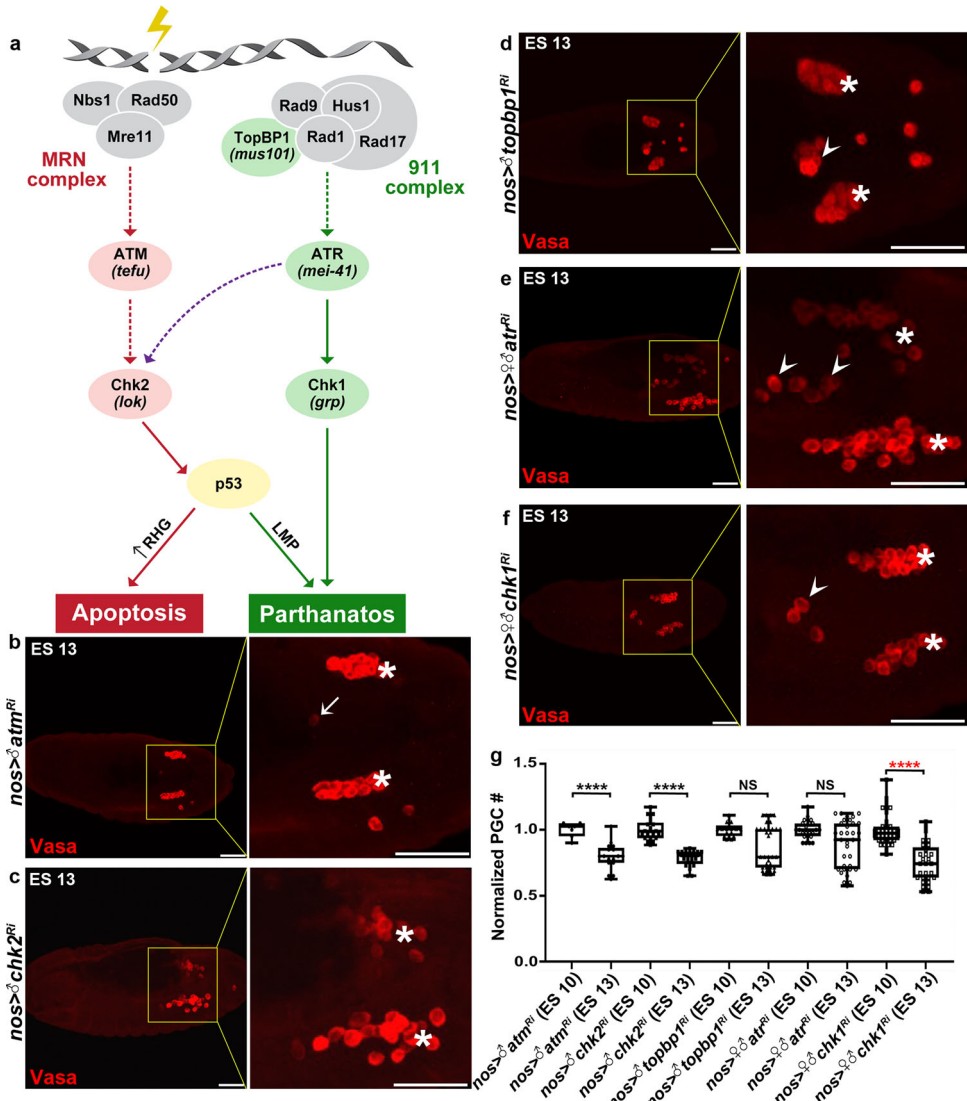

**Fig. 6 Inactivation of the ATR/Chk1 branch of the DDR attenuates PGC death. a** Illustration of the two branches of the DDR pathway induced by DNA damage. Tested components for effects on PGC death are indicated in pink (for no effect) and green (for having an effect). Untested/unconfirmed components are in gray. Dashed lines indicate suggested interactions based on data from mammalian systems or when direct regulation was not observed in *Drosophila*. In brackets are the names of the *Drosophila* genes if different from the names of the mammalian homologs. Note that for the sake of simplicity, only parthanatos is mentioned as a process downstream of Chk1. **b–f** Representative images of ES 13 embryos with PGC-specific knockdowns of the indicated DDR pathway genes (corresponding to the cartoon model in **a**), stained and presented as in Fig. 1b. PGCs (Vasa; red). Dying PGCs are indicated by arrows; ectopically surviving PGCs by arrowheads; gonadal PGCs by asterisks. Whereas PGC death normally proceeded upon knockdown of components in the ATM/Chk2 branch of the DDR (**b**, **c**), knockdown of components in the ATR/Chk1 branch of the DDR attenuated PGC death (**d–f**). Scale bars, 50 μm. **g** Quantifications of PGC death levels in embryos corresponding to the genotypes in (**b–f**). All data points, including outliers, were presented in box plot format where the minimum is the lowest data point represented by the lower whisker bound, the maximum is the highest data point represented by the upper whisker bound, and the center is the median. The lower box bound is the median of the lower half of the dataset while the upper box bound is the median of the upper half of the dataset. Each dot corresponds to the number of PGCs in a single embryo to reflect *n* number, where *n* = number of examined biologically independent embryos. ****$p < 0.0001$; NS, non-significant; Student's *t*-test, one-sided distribution. Added information about the red asterisks is in the main text.

double OE randomly occurs in the entire PGC population (Supplementary Fig. 7e). This is consistent with a recent report suggesting that the (midline) subset of PGCs that undergo cell death is already predetermined at a very early stage of PGC specification, according to the levels of germplasm *wunens* they inherit from the oocyte, which in turn is determined by their spatial position at the posterior pole of the embryo[38]. In contrast to Hid OE induced PGC apoptosis, AIF and DNase II OE induced PGC death was non-apoptotic, as the dying PGCs were TUNEL negative and cell death was not attenuated by OE of the apoptotic inhibitors, Diap1 and Dronc[DN] (Supplementary Fig. 7f–h).

**AIF can bind to DNase II and mediates its nuclear translocation.** Given the involvement of AIF in the nuclear translocation of several nucleases, we next asked whether AIF might mediate PGC death by promoting the nuclear translocation of DNase II. Using the anti-DNase II antibodies to stain *aif* knockdown PGCs, revealed that most of the ectopically surviving midline PGCs in the affected embryos displayed non-nuclear DNase II (Fig. 7e–h). The infrequently observed low levels of nuclear DNase II (in 21.5% of the midline PGCs; Fig. 7h) likely result from incomplete elimination of *aif* by the RNAi transgene. Next, we tested whether AIF can bind to DNase II in vitro by co-expressing full-length

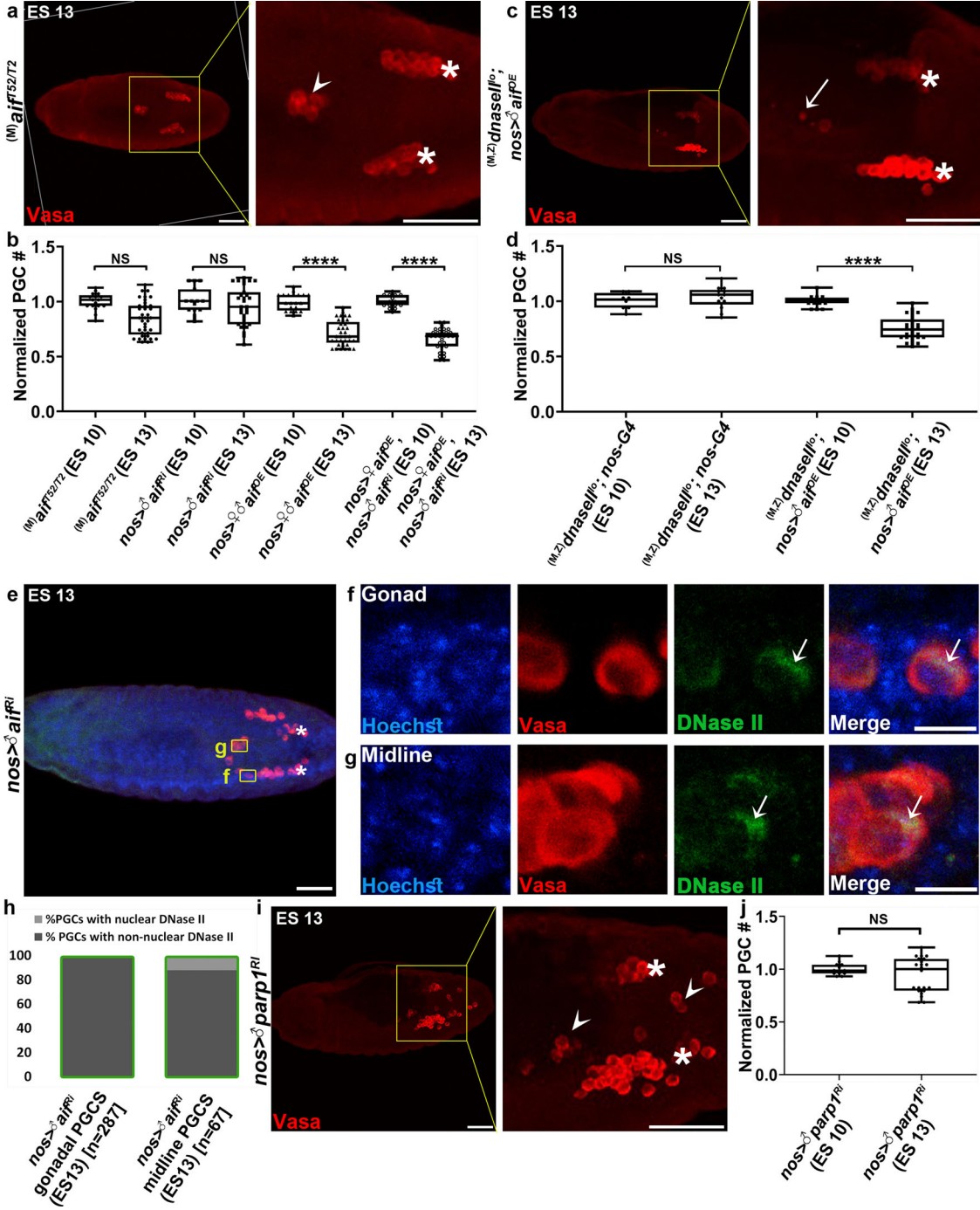

recombinant glutathione S-transferase-tagged AIF (GST-AIF) and FLAG-tagged DNase II (FLAG-DNase II) in *E. coli*. FLAG pulldown analyses from cell lysates revealed that GST-AIF readily associated with FLAG-DNase II, but not with a control recombinant bacterial protein (FLAG-BrxA) or the anti-FLAG beads, indicating that DNase II can physically interact with AIF (Fig. 8a).

**CypA binds to AIF and facilitates DNase II nuclear transloca-tion.** Nuclear translocation and the associated nuclease activity of AIF have been proposed to involve association with cyclophilin A (CypA)[55,60], which similar to other cyclophilins, also contains a peptidyl-prolyl isomerase domain, which can affect protein folding of its associated partners. Searching the *Drosophila*

UniProt for CypA-like protein sequences, revealed 4 different gene products with a cyclophilin-like domain, Cyp1, CG2852, CG17266, and Moca-cyp, sharing 77%, 59%, 57%, and 55% sequence identity with human CypA, respectively. PGC-specific knockdown of *cyp1*, *CG2852*, and *CG17266*, each attenuated PGC death in 60% of the embryos, suggesting an involvement and redundancy among at least three of the CypA-like proteins in PGC death (Fig. 8b, c). To investigate whether the cyclophilins might affect DNase II nuclear translocation, we stained *cyp1* (the most similar *cypA* ortholog) knockdown embryos with anti-DNase II antibodies. Whereas in WT embryos 70% of the dying midline PGCs displayed nuclear DNase II localization, 100% of the ectopically surviving midline PGCs in the *cyp1* knockdown embryos had no nuclear DNase II, suggesting that the cyclophi-lins affect PGC death by modulating (AIF-mediated) DNase II

**Fig. 7 AIF and PARP-1 mediate PGC death. a** Maternally deposited *aif* is required cell autonomously for PGC death. A representative image of an embryo with maternal hypomorphic allele combination of *aif* mutants stained and presented as in Fig. 1b. PGCs (Vasa; red). An arrowhead pointing at ectopically surviving midline PGCs. Asterisks indicate gonadal PGCs. Scale bars, 50 μm. **b** Quantification of PGC death levels in *aif* mutant, knockdown, and overexpressing embryos. All data points, including outliers, were presented in box plot format where the minimum is the lowest data point represented by the lower whisker bound, the maximum is the highest data point represented by the upper whisker bound, and the center is the median. The lower box bound is the median of the lower half of the dataset while the upper box bound is the median of the upper half of the dataset. Each dot corresponds to the number of PGCs in a single embryo to reflect *n* number, where *n* = number of examined biologically independent embryos. ****$p < 0.0001$; NS, non-significant; Student's *t*-test, one-sided distribution. Note that while not inducing precocious PGC death when overexpressed in an otherwise WT background, OE of *aif* was able to oppose the block in PGC death caused by *aif* knockdown, confirming the validity of the *aif* RNAi and OE transgenes. **c, d** *aif* OE restores PGC death levels in embryos maternally mutant for the hypomorphic *dnaseII*[lo] allele. Shown is a representative image of *dnaseII* mutant embryo, at ES 13, with PGC-specific *aif* OE (**c**) stained, presented, and annotated as in Fig. 1b. An arrow is pointing at a condensed dying PGC. Scale bar, 50 μm. Quantification of PGC death levels in the *dnaseII* mutant embryos with or without PGC-specific *aif* OE (**d**), calculated and presented as in (**b**). ****$p < 0.0001$; NS, non-significant; Student's *t*-test, one-sided distribution. **e–g** Nuclear translocation of DNase II is attenuated in ectopically surviving *aif* knockdown PGCs. A representative image of an ES 13 embryo with PGC-specific *aif* knockdown stained to visualize the DNA (Hoechst; blue), PGCs (Vasa; red) and DNase II (green) (**e**), and magnifications of the areas outlined by yellow rectangles (**f, g**). Asterisks indicate gonadal PGCs. Scale bars in (**e**), 50 μm; (**f, g**) 10 μm. **h** Quantification of the percentage of gonadal and midline PGCs with nuclear versus non-nuclear DNase II localization in *aif* knockdown ES 13 embryos. Green column outline indicates that only living cells were counted. *n* number is shown in brackets where *n* = number of examined PGCs. **i, j** PARP-1 inactivation attenuates PGC death. Shown is a representative image of an ES 13 embryo with PGC-specific *parp1* knockdown (**i**) stained, presented, and annotated as in Fig. 1b. Scale bars, 50 μm. Quantification of PGC death levels in embryos with PGC-specific *parp1* knockdown (**j**), calculated and presented as in (**b**). NS, non-significant; Student's *t*-test, one-sided distribution.

nuclear translocation (Fig. 8d, e). Interestingly, whereas PGC-specific OE of AIF in the *cyp1* knockdown embryos restored PGC death in the affected embryos (Fig. 8c), 17% of these embryo displayed excessive PGC death (Fig. 8f), suggesting that the balance between the levels of AIF and the cyclophilins may modulate the cell death promoting activity of AIF. To test whether Cyp1 can indeed physically associate with AIF, full-length recombinant Twin-Strep-tagged Cyp1 (Cyp1-Twin-Strep) and GST-AIF were co-expressed in *E. coli*. Cell lysates used in a Strep-Tactin pull-down analyses showed that whereas Strep-Tactin resin alone could not pull down GST-AIF, the latter was readily co-pulled down with Cyp1-Twin-Strep, demonstrating that Cyp1 can bind to AIF in vitro (Fig. 8g).

**The DNA damage sensor PARP-1 mediates PGC death.** Previous reports have implicated AIF as a key mediator of a related group of non-apoptotic cell death subtypes, commonly termed parthanatos (PAR for Poly(ADP-ribose) and Thanatos, the personification of death in the Greek mythology)[23]. According to the current model, parthanatos is triggered by OE of the DNA damage sensor, Poly(ADP-ribose) polymerase-1 (PARP-1), or its activation following DNA damage caused by genotoxic stress or excitotoxicity, and is manifested by extensive DNA damage[23,28,61]. Given the striking anatomical and molecular similarities between PGC death and parthanatos, we hypothesized that PGC death might constitute a developmental form of this cell death pathway. To address this idea, we first examined whether PARP-1 might also be involved in PGC death. As opposed to mammalian cells, *Drosophila* contains only a single PARP enzyme, Parp1[62]. Critically, PGC-specific *parp1* knockdown attenuated PGC death in 60% of the examined embryos, suggesting a requirement for PARP-1-like activity in PGC death (Fig. 7i, j).

Following recruitment to diverse types of DNA lesions, PARP-1 undergoes conformation shift and activation, promoting the synthesis and attachment of PAR chains to a variety of proteins, hence altering their conformation and structure, as well as facilitating interactions with other proteins. This post-translational modification process is known to regulate a wide variety of cellular processes, including initiation of the DDR and DNA repair pathways[63,64]. To visualize PARP-1 activity during PGC death, we stained ES 10 embryos with anti-PAR antibodies. Nuclear accumulation of PAR was detected in numerous cells in the embryo, with particularly strong expression in some of the midline PGCs (Supplementary Fig. 8a). PGC-specific *parp1* knockdown significantly reduced the PAR signal in these cells, suggesting that most of the PGC PARs are produced by Parp1 (Supplementary Fig. 8b). Therefore, consistent with the genetic data, PARP-1 activity is present in PGCs during the relevant stages of PGC death.

The release of a subset of the AIF protein from the mitochondria and/or its translocation to the nucleus was reported to involve covalent binding of PAR polymers[26]. We therefore reasoned that at least some of the PAR polymers might be detected in the cytoplasmic compartment during PGC death. However, although the prominent nuclear PAR signal disappears during advanced PGC death stages in about 80% of the midline PGCs (while remaining nuclear in 100% of the gonadal PGCs, we could not visualize cytoplasmic accumulation of PAR in the dying midline PGCs (Fig. 9a–d). One explanation for this failure could be that the phase of PGC death involving cytoplasmic PAR activity is very short. To test this idea, we sought to reduce the rate of PGC death by compromising (but not blocking) this ACD pathway. Intriguingly, visualizing PAR in the PGC-specific *chk1* knockdown embryos, where PGC death is attenuated (Fig. 6f, g), revealed ectopically surviving midline PGCs halted at various steps during the release of the PAR polymers to the cytoplasm. Whereas in some of the midline PGCs PAR was still confined to the nucleus (Fig. 9e, g), other PGCs contained both nuclear and cytoplasmic PAR (Fig. 9e, h), while a third PGC population displayed PAR exclusively in the cytoplasm (Fig. 9e, i). As expected, in the gonadal PGCs, PAR exclusively resided in the nucleus (Fig. 9e, f). Taken together, these observations are in line with the parthanatos model in which nuclear PAR generated by PARP-1 is released and signals in the cytoplasm.

**DNase II and PARP-1 engage in positive amplification loop.** Our results thus far support a model in which PGC death is mediated by DNase II-induced DNA damage and by PARP-1-dependent activation of the DDR (see the cartoon model in Fig. 10). To investigate the inter-regulation between these two modalities in PGC death, we monitored PARP-1 activity (the presence of PAR polymers) in PGCs of *dnaseII*[lo] mutant embryos, as well as the nuclear translocation of DNase II in the background of PGC-specific *parp1* knockdown. In contrast to the dying PGCs in WT embryos, in which an intense nuclear PAR signal

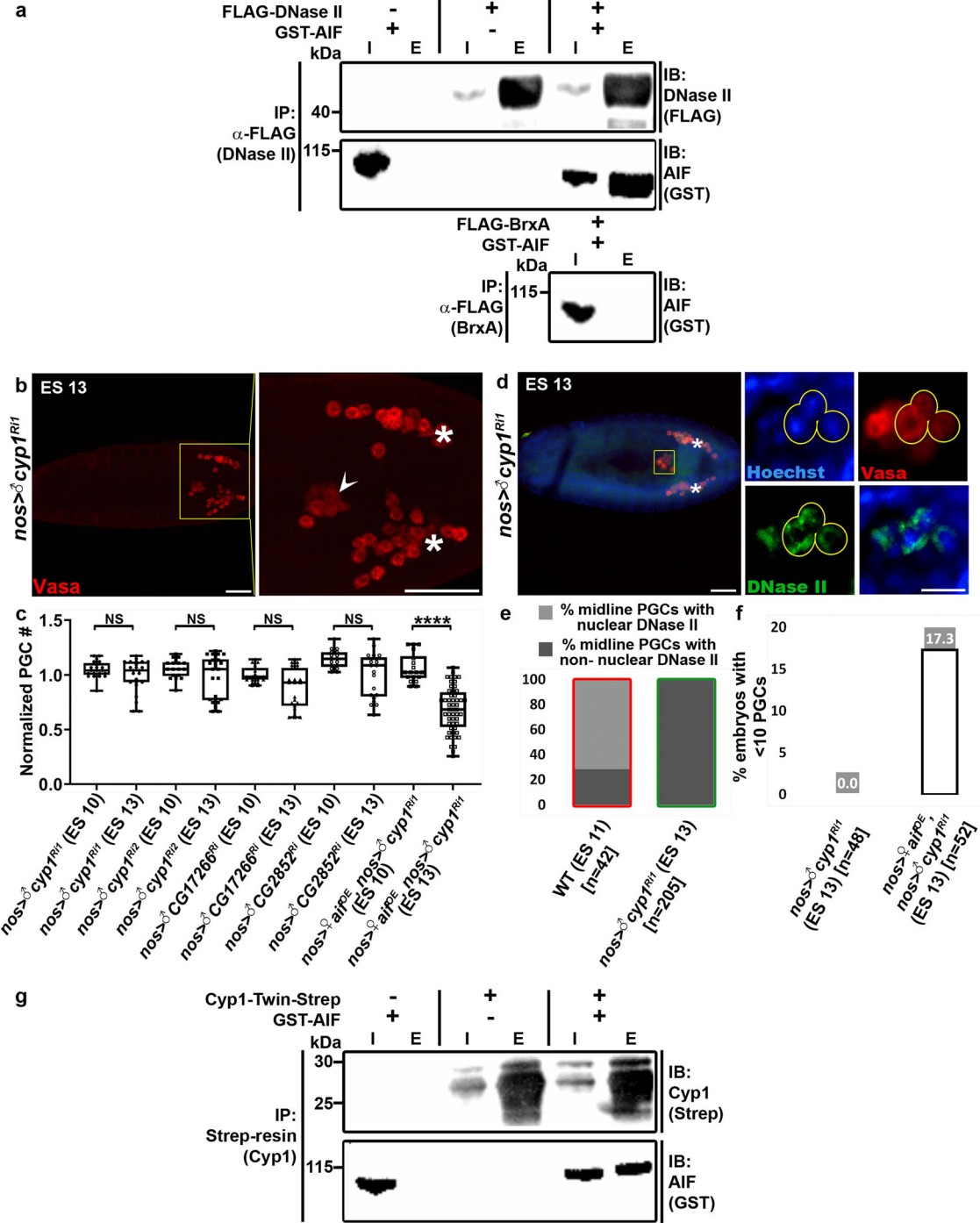

disappears during advanced PGC death stages (Fig. 9a-c), ectopically surviving midline PGCs in the *dnaseII^{lo}* mutants displayed persistent nuclear PAR signal, suggesting that the release of the PAR polymers to the cytoplasm requires DNase II activity (Supplementary Fig. 8c-e). Conversely, when *parp1* was knocked down, the translocation of DNase II to the nucleus in the ectopically surviving midline PGCs was significantly reduced (detected only in two PGCs within two different embryos out of 50 examined embryos), suggesting that PARP-1 activity modulates DNase II nuclear translocation (Supplementary Fig. 8f-i). Given the synergistic interaction between AIF and DNase II, and that AIF modulates the nuclear translocation of DNase II during PGC death, as well as the idea that PARP-1 activity (cytoplasmic PAR) promotes the release of AIF from the mitochondria and translocation to the

nucleus[26], these observations suggest that PARP-1 and DNase II engage in a positive feedback amplification loop, leading to activation of the DDR and consequent PGC death by parthanatos.

## Discussion

In this study, we discovered a developmental form of the ACD pathway called parthanatos, by which PGCs undergo cell death during development, demonstrating that parthanatos is not limited to stress or pathological conditions. Our collective results support a model in which developmental parthanatos is triggered by lysosomal components and is mediated by the DDR. At the center of this pathway is a positive feedback amplification loop involving PARP-1 and DNase II, which through PAR and AIF mediators, enhances PARP-1 activation and DNase II nuclear translocation, respectively,

**Fig. 8 Interplay between AIF, CypA, and DNase II. a** AIF physically associates with DNase II in vitro. Recombinant DNase II and AIF (tagged with FLAG and GST, respectively) were expressed alone or together in *E. coli*. AIF was pulled down by immunoprecipitation of DNase II with anti-FLAG antibodies, and the presence of AIF was confirmed using anti-GST antibodies. As a negative control, FLAG-tagged BrxA (a *Bacillus cereus* protein) was similarly expressed with GST-AIF and pulled down with anti-FLAG antibodies. Both the immunoprecipitates (E, elution) and the corresponding preincubated lysates (I, input) were analyzed by immunoblotting (IB). Note the potent binding of GST-AIF to FLAG-DNase II but not to FLAG-BrxA or to the anti-FLAG beads. **b, c** The *Drosophila cypA* orthologs are involved in PGC death. Shown is a representative image of an ES 13 embryo with PGC-specific knockdown of the closest *cypA* ortholog, *cyp1* (**b**), stained to visualize the PGCs (Vasa; red) and presented as in Fig. 1b. An arrowhead pointing at ectopically surviving midline PGCs. Asterisks indicate gonadal PGCs. Scale bars, 50 μm. The corresponding quantifications of *cyp1* knockdown embryos, and embryos with knockdowns in the other *cypA* orthologs are shown in (**c**). All data points, including outliers, were presented in box plot format where the minimum is the lowest data point represented by the lower whisker bound, the maximum is the highest data point represented by the upper whisker bound, and the center is the median. The lower box bound is the median of the lower half of the dataset while the upper box bound is the median of the upper half of the dataset. Each dot corresponds to the number of PGCs in a single embryo to reflect *n* number, where *n* = number of examined biologically independent embryos. ****$p < 0.0001$; NS, non-significant; Student's *t*-test, one-sided distribution. **d, e** Nuclear translocation of DNase II is blocked in ectopically surviving midline PGCs in *cyp1* knockdown embryos. Shown is a representative image of an ES 13 embryo with PGC-specific *cyp1* knockdown (**d**) stained and presented as in Fig. 3k, l. Asterisks indicate gonadal PGCs. Scale bars, 50 μm. Note the non-nuclear DNase II localization in the ectopically surviving midline PGCs (yellow circles). Scale bars, 10 μm. **e** Quantification of the percentage of midline PGCs with nuclear and non-nuclear DNase II localization for embryos of the indicated genotypes and ES. Green column outline indicates that the cells being counted are living PGCs, red column outline indicates that the cells being counted are dying PGCs. *n* number is shown in brackets where *n* = number of examined PGCs. **f** Quantification of the percentage of ES 13 embryos with less than 10 PGCs following PGC-specific OE of *aif* in embryos with PGC-specific *cyp1* knockdown. The percent value is indicated above each column. *n* number is shown in brackets where *n* = number of examined biologically independent embryos. Note that while OE of *aif* restored PGC death in the *cyp1* knockdown embryos (**c**), in about 17% of the embryos it caused excessive PGC death (**f**). Since OE of *aif* in an otherwise WT background does not affect PGC death levels (Fig. 7b), this finding suggests that the relative Cyp1 levels may modulate the AIF potency to promote cell death. **g** AIF physically associates with Cyp1 in vitro. Recombinant Cyp1 and AIF (tagged with Twin-Strep and GST, respectively) were expressed alone or together in *E. coli*. AIF was pulled down by co-purification of Cyp1 with Strep-Tactin, and the presence of AIF was confirmed using anti-GST antibodies. Both the immunoprecipitates (E, elution) and the corresponding preincubated lysates (I, input) were analyzed by immunoblotting (IB). Note the potent binding of GST-AIF to Cyp1-Twin-Strep but not to Strep-Tactin resin.

culminating in the activation of the ATR/Chk1 branch of the DDR pathway and consequent cell death (Fig. 10). This model significantly expands the current knowledge about parthanatos, as it implicates both lysosomal components, in particular DNase II, and the DDR in the activation and execution of this ACD pathway, respectively. Furthermore, contrary to the notion that DNA fragmentation/damage constitutes the final/execution stage of parthanatos, the current study demonstrates that activation of the DDR pathway downstream of DNA damage is essential for the execution of this ACD pathway.

Molecularly, parthanatos is mediated and hence defined by the action of 3 critical components: PARP-1, AIF, and a PARP-1-dependent AIF-associated nuclease (PAAN)[27,28,65]. Although additional comparative studies of the current parthanatos paradigms, as well as of yet unidentified parthanatos subtypes, are still required in order to reliably generalize a common mechanism, it appears that PARP-1 and AIF are invariably required for triggering and mediating parthanatos, while the identity of the PAAN could vary between different systems and cell types. Whereas in the current work we identified DNase II as the AIF interacting nuclease critical for PGC death, a recent study in mammalian cells revealed MIF, which does not have a *Drosophila* homolog, to be the critical PAAN[27]. In addition, AIF was shown to interact with other nucleases, such as EndoG and LEI/LDNaseII, promoting apoptosis in *C. elegans* and triggering caspase-independent cell death (presumably parthanatos) in mammalian cells, respectively[57,66]. We therefore propose that the exact identity of the nuclease involved might serve as an initial basis for classification of the different parthanatos subtypes. Furthermore, the remarkable anatomical and molecular conservations between PGC death (DNase II-parthanatos) and MIF-parthanatos might also imply that, similar to DNase II-parthanatos, mediation and execution of other parthanatos subtypes may also involve the DDR.

Our findings that inactivation of DNase II partially rescued PGC death induced by the lack of maternal *wunens*, place DNase II-parthanatos pathway downstream of Wunens in the PGCs. How the Wunens trigger PGC death is unclear, but it was suggested that the balance between the soma and germline expressed

Wunens controls PGC survival and death, by competing for the hydrolysis of an extracellular lipid phosphate[36,37]. Interestingly, Wun2 was shown to also promote internalization and rapid cytoplasmic accumulation of the dephosphorylated lipid substrates[36]. Given that chronic lipid overload can promote lysosome dysfunction and LMP[67], a conceivable hypothesis would be that the induction of midline PGC death might occur by Wunens-dependent lipotoxicity-induced LMP.

In conclusion, cell death of the aberrantly migrating cells are evolutionary conserved features of PGCs[34]. Intriguingly, PGC death in mice might also diverge from canonical apoptosis: although mouse embryos lacking Bax, the proapoptotic Bcl-2 family member, display a delay in PGC death, the ectopically surviving PGCs still die via a Bax-independent mechanism later in development, implying the involvement of an ACD pathway[68]. Furthermore, a caspase-independent role of Bax in lipid-induced LMP was also noted[69], suggesting that similar mechanisms might operate in both *Drosophila* and mouse PGCs to trigger cell death. Future comparative studies of PGC death in *Drosophila* and mammals, as well as of other parthanatos paradigms, will improve insight into the signaling pathways and mechanisms underlying this ACD pathway, and will shed light on the molecular commonalities and differences between the different parthanatos subtypes and their possible significance. Since cell death by parthanatos has been implicated in the pathogenesis of many important human diseases[29–31], and because targeted induction of parthanatos could overcome the inherent resistance of many cancer cells to (apoptotic) cell death, the importance of addressing these questions is of considerable significance for translational research and applications.

## Methods

***Drosophila* strains and crosses.** Flies were raised on standard yeast/molasses medium at 25 °C. The following stocks were used: Oregon R and *nos-Gal4-VP16* (on 2nd chromosome; was used in experiments shown in Figs. 6f, 7c, 9e, Supplementary Fig. 1k, Supplementary Fig. 4e) from L. Gilboa (Weizmann Institute, Israel); Canton S, UAS-*myc-diap1* and UAS-*dronc*DN from H.D. Ryoo (NYU School of Medicine, NY); *nos-Gal4-VP16* (on 3rd chromosome; Bloomington *Drosophila* Stock Center [BDSC], stock #4937; was used in all the relevant experiments except for the ones mentioned above); UAS-*p35* and UAS-*hid7* from

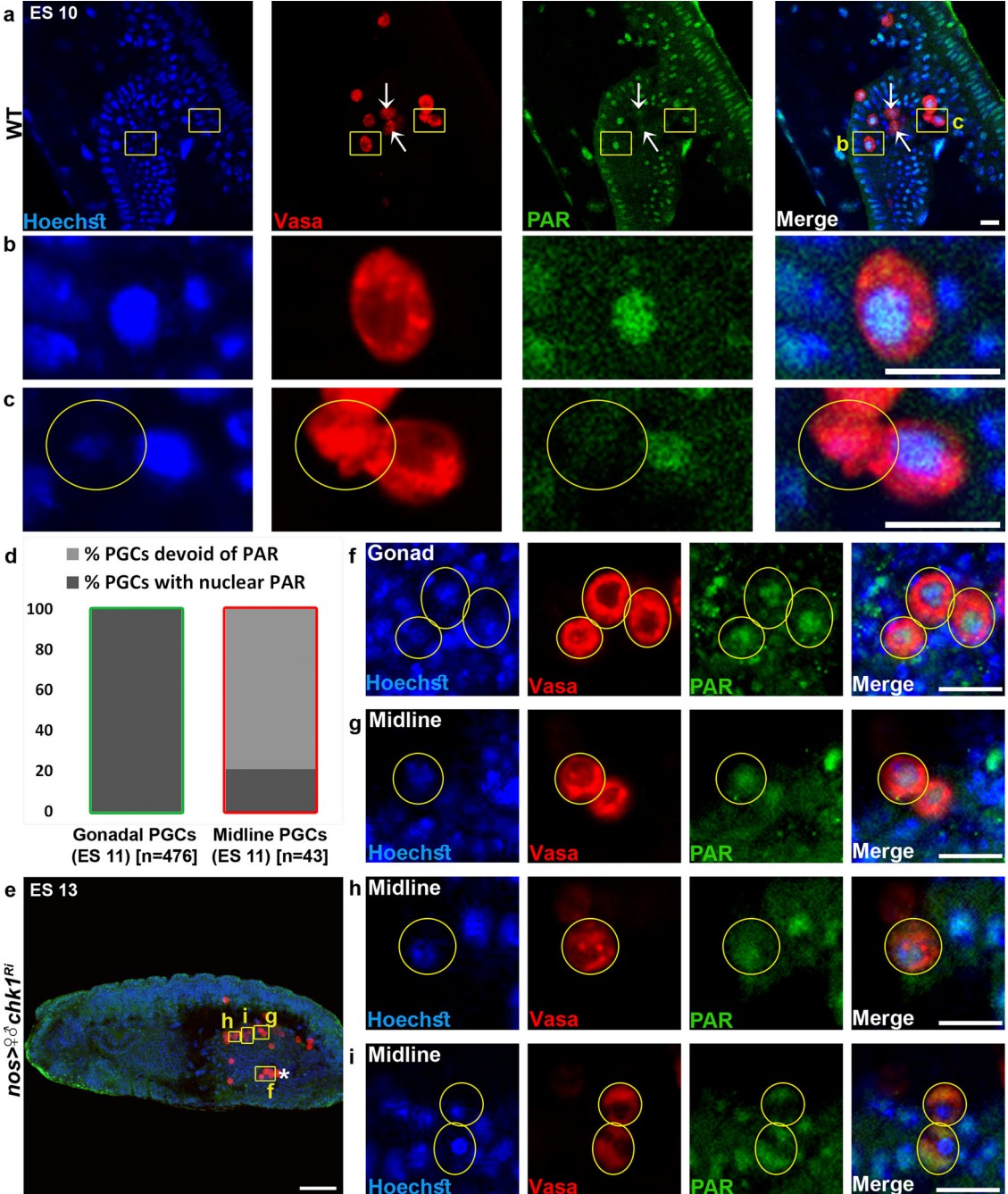

**Fig. 9 PAR polymers are released to the cytoplasm during PGC death. a–d** Dying PGCs display no PAR polymers in the nucleus. A representative image of the midline region of an ES 10 WT embryo stained to visualize PAR (anti-PAR; green), PGCs (Vasa; red), and DNA (Hoechst; blue) (**a**), and magnifications of the areas outlined by yellow rectangles (**b**, **c**). Arrows are pointing at two dying PGCs with highly reduced levels of nuclear PAR. Note that the PAR polymers are predominantly localized in the nucleus of living (intact) PGCs, but are almost absent in dying PGCs (a condensed dying PGC is circled). Scale bars, 10 µm. Quantification of the percentage of gonadal and midline PGCs devoid of PAR signal or with nuclear PAR in WT ES 11 embryos (**d**). Green column outline indicates intact living cells, while red column outline indicates condensed and distorted dying PGCs. *n* number is shown in brackets where *n* = number of examined PGCs. **e–i** Ectopically surviving PGCs due to compromised DDR reveal the release of the PAR polymers from the nucleus to the cytoplasm. Shown is a representative image of an ES 13 embryo with PGC-specific *chk1* knockdown (**e**) stained as in (**a**). An asterisk indicates gonadal PGCs. Magnifications of several areas outlined by yellow rectangles in (**e**) are presented in (**f–i**). Shown are gonadal PGCs (**f**) and an ectopically surviving midline PGC (**g**) predominantly displaying PAR in the nucleus (circled), an ectopically surviving midline PGC displaying both nuclear and cytoplasmic PAR (**h**; circled), and ectopically surviving midline PGCs displaying only cytoplasmic PAR (**i**; circled). Scale bars in **e** 50 µm, **f–i** 10 µm.

H. Steller (Rockefeller University, NY); *drice*[Δ1];[70] *dcp-1*[prev];[71] *strica*[4];[72] *dredd*[B118] (BDSC Stock #55712);[73] *damm*[f02209] (BDSC Stock #18524);[74] *dnaseII*[lo];[59] *cathD*[1];[75] *drpr*[Δ5];[43] UAS-*puc*;[76] *atg7*[d77] and *atg7*[d14];[77] *atg1*[KQ38A];[78] *omi*[Δ1] ;[79] *endoG*[MB07150];[80] UAS-*lamp-GFP* from E. Baehrecke (UMass Medical School, MA); UAS-*hsp70*;[81] UAS-*orc2* (FlyORF #F000944)*; UAS-*bub3* (FlyORF #F000892)*; *p53*[5A-1-4];[82] *aif*[52] (VDRC JP Stock, #311002);[83] *aif*[T2] (VDRC JP Stock, #311004);[83] UAS-*aif* (VDRC

JP Stock, #311001);[83] *70I-CreI*;[51] *wun2*[N14] and *wun*[Df];[35] *matα-GAL4* (BDSC Stock #7063)[84].

RNAi lines against the following genes are from the TRiP collection at Harvard Medical School and were obtained from the BDSC: *egfp* (#41556); *dredd* (#34070);[85] *dnaseII* (#63635)*; *dor* (#44426)*; *car* (#34007);[86] *parp1* (#57265);[87] *cathB* (#33953);[88] *cathL* (#41939);[89] *atm* (#44417)*; *chk2* (#64482);[90] *atr*

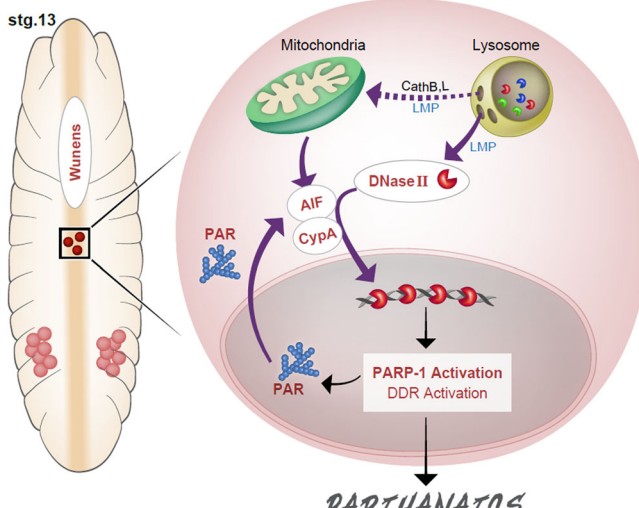

**Fig. 10 An integrated model of the developmental form of PGC death by parthanatos.** An illustration of an embryo at ES 13 (left) and an enlargement of a dying PGC (right), featuring the molecular model of PGC death. Purple arrows indicate translocations from one cellular compartment to another, while black arrows indicate signaling. PGCs that remain in the midline experience high somatic Wunens (LPP) activity, which may induce LMP and the release of lysosomal hydrolases, including DNase II, CathB and CathL (See the Discussion for more details about a possible link between the Wunens and LMP). The release of CathB, L from the lysosomes might promote the initial release of AIF from the mitochondria as was reported[100] (dashed arrow). AIF then, together with CypA, promotes the translocation of DNase II to the nucleus, presumably as a CypA-AIF-DNase II complex. In the nucleus, DNase II cleaves the DNA, leading to PARP-1 activation, which in turn generates PAR. PAR modifies proteins both in and outside the nucleus, activating the DDR and promoting massive release of AIF from the mitochondria, respectively. This positive feedback amplification loop between DNase II and PARP-1 (mediated by AIF and PAR, respectively), activates the ATR/Chk1 branch of the DDR (presumably to a critical threshold), ultimately leading to PGC death.

(#41934);[90] chk1 (#36685);[90] atg7 (#34369);[91] atg6 (#35741);[91] atg5 (#34899);[92] atg8a (#34340);[92] duox (#32903);[93] orc2 (#43215)*; cyp1$^{Ri1}$ (#33950);[94] cyp1$^{Ri2}$ (#33001)*; CG17266 (#58350)*; CG2852 (#55196)*.

RNAi lines against the following genes were obtained from the Vienna *Drosophila* RNAi Center (VDRC): decay (KK100168);[95] p53 (GD45139); aif (KK109615)*; topbp1 (GD31431)*.

The UAS-dnaseII transgenic fly line was generated by obtaining a pUAST-BD vector containing the *dnaseII* gene from the DRSC pUAST-BD ORF collection (Clone ID: DmCD00356137). The transgenic flies were generated by micro-injection into embryos performed by BestGene Inc.

*The fly lines marked with asterisks were validated in the current study for maternal overexpression or knockdown of the indicated genes (Supplementary Fig. 9), and in genetic rescue experiments. All other lines were either validated in previous studies from our laboratory or generated and validated in the corresponding citations.

For PGC transgenic expression, females (♀) carrying the maternal driver *nos-Gal4-VP16*, were crossed with males (♂) carrying the different *UAS-*dependent transgenes. Target *UAS-*based transgenes of this potent Gal4 driver begin to be expressed in PGCs at ES 8-9, prior to their migration from the posterior midgut[42]. The examined embryos carried either single copies of the driver (*nos*) and the transgene (♂), or two copies of each of these elements (i.e. when a stable fly line was generated; ♀, ♂). For the mutant embryos, we usually generated two genetic constellations. Crossing homozygous mutant females with WT males produced maternally mutant embryos (M), whereas crossing between heterozygous mutant females and males gave rise to zygotically mutant embryos (Z). Embryos that are both maternally and zygotically mutant (M, Z) were produced by a stable homozygous mutant fly line.

**Immunofluorescence staining and visualization.** Flies of the appropriate genotype were allowed to lay eggs on juice agar plates with a dollop of thick yeast paste for two hours at 25 °C. Embryos were then aged for 3.5 h at 25 °C to reach ES 10, 5.5 h at 25 °C to reach ES 11, 15 h at 18 °C to reach ES 12, 17 h at 18 °C to reach ES 13, 19.5 h at

18 °C to reach ES 14, 23 h at 18 °C to reach ES 15, 14 h at 25 °C to reach ES 16, and 16 h at 25 °C to reach ES 17. For staining with anti-Vasa antibody, staged embryos were collected in mesh baskets, dechorionated in 50% bleach, washed with 1× PBS + 0.1% TritonX-100 (PBTx), and fixed in 4% formaldehyde, 1.75 ml PEMS (100 mM PIPES pH 6.9, 1 mM EGTA, 2 mM MgSO₄), and 8 ml heptane for 20 min. Vitelline membranes were removed by manual shaking in 100% methanol for 1 min.

For staining with multiple antibodies, we used an alternative fixation as follows: Staged embryos were placed in boiling Triton/Salt (0.7% NaCl, 0.04% TritonX-100) solution for 5 s then moved to cool Triton/Salt solution on ice for 15 min. Vitelline membranes were removed by manual shaking in 50% methanol:50% heptane for 1 min.

Embryos were rehydrated, washed with PBTx, and blocked with 5% normal goat serum in PBTx. The embryos were then incubated in primary antibodies overnight at 4 °C, washed again in PBTx and incubated with secondary antibody for two hours at room temperature. When applicable, embryos were placed in 1:500 Hoechst 33342 (H3570, Invitrogen):PBTx for 2 min. Embryos were washed in PBTx before being mounted onto slides in Fluoromount medium (SouthernBiotech, Birmingham, AL, USA) and were visualized using a confocal microscope (Zeiss LSM710, LSM780 and LSM800) or a Dragonfly spinning disc microscope for high resolution images.

For Supplementary Movie 1, Z sections of 0.2 μm each (57 sections in total) were taken. Z stack series were combined into a movie using Surfaces technology in the Imaris software (version 9.6.0).

**Antibodies.** The primary antibodies used in this study were polyclonal rabbit anti-Vasa (1:250, sc-30210; Santa Cruz), mouse anti-PAR (1:20, ab14459; Abcam), rabbit anti-cleaved Dcp-1 (1:100, 9578 S; Cell Signaling), rabbit anti-Hb9 (1:100; from J. Skeath, Haverford College, PA), and the following antibodies from the Hybridoma Bank (DSHB): polyclonal rabbit anti-Arl8 (1:20) and monoclonal mouse anti-Vasa (direct, 46F11) and mouse anti-γH2Av (direct, UNC93-5.2.1).

Anti-DNase II antibodies were generated as follows: GST-tagged DNase II protein without the signal peptide (spanning amino acids 1-19) was expressed in E. coli. A Coomassie Blue-stained gel slice of the corresponding size was used as an antigen to generate polyclonal antibody in two rats. One immunization and four boosts were done for each animal and after the 4th boost 60 ml serum were obtained. The antibody was used at a dilution of 1:20.

**TUNEL and Top I-mediated ligation assay.** Embryos were staged, collected, and fixed as described above. TUNEL labeling was carried out using the ApopTag Red In Situ Apoptosis Detection Kit (Millipore, Billerica, MA, USA). Fixed embryos were washed in 1× PBS + 0.1% Tween 20 (PBTw), washed in equilibration buffer for one hour at room temperature, incubated in TdT reaction mix (70% reaction buffer to 30% TdT enzyme) for three hours at 37 C°, and then incubated in stop buffer for four hours at 37 C°. Next, embryos were blocked in BTN solution (1xBSS, 0.3% Triton X-100, and 5% normal goat serum) for one hour at room temperature, washed in 1xBSS, incubated in anti-dioxigenin and primary antibody overnight at 4 °C, washed with 1xBSS, incubated in secondary antibody for two hours at room temperature, washed, mounted and visualized as described above.

Top I-mediated ligation assay was carried out using the ApopTag ISOL Dual Fluorescence Apoptosis Detection Kit (DNase Types I & II) (Millipore, Billerica, MA, USA). Fixed embryos were rehydrated, washed with PBTw, incubated in the ISOL labeling reaction mixture in a dark humid chamber for 16 hours at 18 °C, and then blocked in BTN solution for one hour at room temperature. Embryos were then incubated with primary antibody overnight at 4 °C, washed with 1xBSS, incubated with secondary antibody for 2 hours at RT, washed, mounted, and visualized as described above.

**Quantification of PGC death.** The number of Vasa-positive PGCs in each embryo was manually counted following confocal microscope imaging. For each genotype, the average PGC number at ES 10 was set as 1 and the individual values at ES 10 and 13 were expressed as a percentage of the average value at ES 10. The values were then plotted using box and whisker charts, in which the median is shown as a line in the middle of the box and the minimum and maximum values are the bottom and the top of the whiskers respectively. Each dot in the box represents a single embryo clearly reflecting the sample number (n). On average, a sample number of 23 was used for each genotype. Note, whereas at ES 13 most of the dead PGCs are already eliminated, dying PGCs can be sometimes still detected by virtue of their highly condensed morphology and relatively faint Vasa staining signal, and are omitted from the counts.

PGC death index in Fig. 1d was calculated as follows: The number of embryos with at least one dying PGC (condensed/distorted) was divided by the total number of examined embryos giving a value A. The total number of dying PGCs in all the embryos was divided by the number of embryos with at least one dying PGC, giving a value B. Death index equals AxB.

**Protein co-purification.** Cloning: All cloning reactions were performed by the Restriction-Free (RF) method[96]. Full length open reading frame (ORF) of *Drosophila aif* was amplified from a cDNA clone (AT03068) and sub-cloned into the expression vector pETHGST, a modified version of pETGST-TevH[97]. Full length ORFs of *Drosophila dnaseII* (cDNA clone GH10876) and *Bacillus cereus brxA* (a

gift from Rotem Sorek, WIS) genes were sub-cloned into the 1st ORF of the expression vector pACYCDuet-1 (Novagen), including an N-terminal FLAG-tag followed by a TEV cleavage site. Full-length *Drosophila* ORF of *cyclophilin 1* (*cyp1*; cDNA clone RE62690) was sub-cloned into the 2nd ORF of pACYCDuet-1, including a C-terminal Twin-Strep-tag.

Protein expression: Individual protein expression and co-expression studies were performed in *E. coli* BL21(DE3) cells. Expression was performed in LB medium supplemented with the appropriate antibiotics (Kanamycin and/or chloramphenicol). Expression was induced with 200 μM IPTG followed by shaking at 15°C for ~16 h. Cell pellets were stored at -20 °C before processing.

Protein pulldown: For FLAG-tag pulldown, cells were lysed by sonication in Tris-buffered saline (TBS) buffer supplemented with 1 mM phenylmethylsulfonyl fluoride (PMSF) and 1 μl/mL of protease inhibitor cocktail (Set IV, EMD Chemicals, Inc). Protein pulldown experiments were performed using Anti-DYKDDDDK G1 affinity Resin (GenScript #L00432,) according to the manufacturers' recommendations. Western blotting was performed using mouse Anti-Flag-tag [HRP] mAb (GenScript # A01428-100) and mouse Anti-GST [HRP] mAb (GenScript #A00866-100). Dilution used for all antibodies was 1:1000.

For Strep-tag pulldown cells were lysed by sonication in 50 mM Tris-HCl, pH 8, 500 mM NaCl buffer supplemented with 1 mM phenylmethylsulfonyl fluoride (PMSF) and 1 μl/mL of protease inhibitor cocktail (Set IV, EMD Chemicals, Inc). Protein pull-down experiments were performed using Strep-Tactin Sepharose resin (#2-1201-010, IBA) according to the manufacturers' recommendations. Western blotting was performed using THE™ NWSHPQFEK Tag Antibody [HRP], mAb, Mouse (GenScript #A01742-40) and mouse Anti-GST [HRP] mAb (GenScript #A00866-100). Dilution used for all antibodies was 1:1000.

**Validation of knockdown and overexpression fly lines**. Western blotting: Fifty testes were dissected from young adult WT and *bam-Gal4 > aif^{Ri}* males, and were homogenized and sonicated in 50 μl of RIPA buffer (10 mM Tris-HCl, pH 8.0, 1 mM EDTA, 0.5 mM EGTA, 1% Triton X-100, 0.1% Sodium Deoxycholate, 0.1% SDS, and 140 mM NaCl), to prepare protein extracts. The supernatant protein concentrations were determined and 50 μg from each sample were loaded on an SDS-PAGE gel. The proteins were transferred from the gel onto a nitrocellulose membrane, which after blocking in 5% milk solution (dry milk in PBTw) was incubated with the primary antibody (anti-AIF antibody), followed by anti-rat HRP secondary antibody for one hour at room temperature. After three washes in PBTw, images were obtained using the ImageQuant LAS 4000 mini. The membrane was then stained with Ponceau S for loading control.

Anti-*Drosophila* AIF antibody was generated as follows: GST-tagged AIF protein segment spanning the predicted FAD-dependent oxidoreductase domain (amino acids 257–564) was expressed in E. coli. A Coomassie Blue-stained gel slice of the corresponding size was used as an antigen to generate polyclonal antibodies in two rats. One immunization and four boosts were performed in each animal and after the 4th boost, 60 ml serum was obtained. The antibody was used at a dilution of 1:500.

RNA isolation and real-time PCR analysis: Successful knockdowns of the indicated genes were validated by comparing the expression level of each gene between the control embryos (carrying the maternal driver *matα-GAL4*) and embryos carrying both the *matα-GAL4* and the indicated *UAS-RNAi* construct. Fertilized *Drosophila* eggs were collected for 30 min after egg-laying, and total RNA was isolated using the Quick-RNA Microprep Kit (Zymo Research, #R1051). cDNA was then synthesized using the High-Capacity cDNA Reverse Transcription Kit (Applied Biosystems™ #4368814). Gene expression levels were determined by real-time PCR using thr StepOnePlus™ Real-Time PCR System (Applied Biosystems™ #4376600) with the KAPA SYBR® FAST qPCR Master Mix Kit (KAPA Biosystems #KR0389_S-v2.17). Expression values were normalized to α-Tubulin at 84B (αTub84B, CG1913). Fold change in the expression levels were analyzed using the ΔΔCt method[98]. Primer sets were designed using the FlyPrimerBank online database[99]and are listed herein.

| | | | |
|---|---|---|---|
| *tub84B* (CG1913) | Forward | GATCGTGTCCTCGATTACCGC | |
| | Reverse | GGGAAGTGAATACGTGGGTAGG | |
| *orc2* (CG3041) | Forward | TGGTTGGGAATGCAGTGGAAT | |
| | Reverse | CGCCTGGTATTTCTTGGTGGG | |
| *bub3* (CG7581) | Forward | TTGGGACGGAACACTTAGATTCT | |
| | Reverse | CTATGTCCATGAAGGCACAGTC | |
| *dor* (CG3093) | Forward | GGAGCGACAACTCTGCTG | |
| | Reverse | GCGTGTTATTTTATAGCCGGAGC | |
| *atm* (CG6535) | Forward | AGCGGAGGTCATTCATATCG | |
| | Reverse | TGTTAAACGAAAGGGAACGG | |
| *topbp1* (CG11156) | Forward | CTGATCGCAGCCGGTAACC | |
| | Reverse | CATGGCCGTGCTGTAGATG | |
| *cyp1* (CG9916) | Forward | ATTTCAGATTGCGGAGTC | |
| | Reverse | TATTCACACTGCACGCTGACG | |
| CG17266 | Forward | AGAAGCTCCAACAATCCCGTC | |
| | Reverse | TTGTAGCCAATGGGAACGCC | |
| CG2852 | Forward | AAGCTGTTCTTATCCGTTTTCGT | |
| | Reverse | CTTTGGGACCCTTGCTATCGT | |

**Statistics and reproducibility**. Statistical significance was calculated using a one-tailed homoscedastic Student's *t*-test or Fisher's test. *P*-values are indicated in figures and in figure legends. Error bars indicate SD. For all statistical analyses values of *P* < 0.0001 were accepted as statistically significant, except for supplementary Fig. 9, where the *P*-values are defined in the relevant legend.

For all experiments presented as representative images, 5-23 biological replicates were performed except for the following: Figs. 4d, 8a, Supplementary Figs. 4e and 9c, two biological replicates each; Figs. 5e, 8g, and 9e–i, one biological replicate each; Supplementary Fig. 9d–k, three biological replicates each.

We have complied with all relevant ethical regulations for animal testing and research, including invertebrates.

**Reporting summary**. Further information on research design is available in the Nature Research Reporting Summary linked to this article.

## Data availability
The authors declare that the data supporting the findings of this study are available within the paper and its supplementary information files. All unique materials generated are readily available from the authors. Source data are provided with this paper.

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

## Acknowledgements

We are grateful to Lilach Gilboa, Hyung Don Ryoo, Hermann Steller, Bruno Lemaitre, Joseph Penninger, Eric Baehrecke, Jim Skeath, Akira Nakamura, the Vienna *Drosophila* RNAi Center (VDRC), the *Drosophila* RNAi Screening Center (DRSC) and Transgenic RNAi Project (TRiP), and the Bloomington *Drosophila* Stock Center for providing additional stocks and reagents. We thank the Arama laboratory members for their encouragement and advice. We note Asa Tirosh and Dr. Shira Albeck from the Israel Structural Proteomics Center (ISPC) at the WIS for helping to carry out the co-immunoprecipitation experiments. We warmly thank Genia Brodsky from the WIS Graphic Design Department for help with the graphic illustrations. We warmly thank Eyal Schejter for his excellent comments on the manuscript. This research was supported by grants from the European Research Council under the European Union's Seventh Framework Programme (FP/2007-2013)/ERC grant agreement (616088), the ISRAEL SCIENCE FOUNDATION (grant No. 1279/19), and the Minerva Foundation with funding from the Federal German Ministry for Education and Research. E.A. is supported by a research grant from the Estate of Emile Mimran. E.A. is the Incumbent of the Harry Kay Professional Chair of Cancer Research.

## Author contributions

E.A. and L.T-I. designed the project and analyzed all experiments. Execution of the experiments was carried out by all the authors. E.A. and L.T-I. wrote the manuscript. All authors discussed the results and commented on the manuscript.

## Competing interests

The authors declare no competing interests.
