## [Peer Review File · Nature Communications]

REVIEWER COMMENTS

Reviewer #1 (Remarks to the Author):

This article explores the mechanisms by which primordial germ cells (PGCs) in *Drosophila*, which failed to reach the somatic gonad, die. The authors provide an interesting link to a nonapoptotic cell death pathway, called Parthanatos, which is characterized by accumulation of Par and nuclear translocation of AIF. Several previous studies in *Drosophila* (Nakamura, Lehmann, Coffman labs) have demonstrated that PGCs, which do not reach the gonad, die and that this death is non-apoptotic. The regulation of germ cell death during *Drosophila* embryogenesis has been attributed to two pathways, the lipidphosphate phosphatase Wunen, which acts both non-autonomously and germ cell autonomously and a P53 dependent pathway that acts independently of Wunen/Wunen2 (Starz-Gaiano et al 2001; Renault et al 2004, 2010; Hanyu-Nakamura et al 2004, Yamada et al 2008, Slaidina et al 2017). The present manuscript presents new evidence regarding the cellular mechanism of this cell death, however how the parthanatos pathway links to these known pathways, which have been shown to trigger PGC death, needs to be addressed. Furthermore, throughout the manuscript many components are identified that affect PGC death, in some case a physiological role for these genes in PGCs need to be demonstrated, for others a link to the AIF/DNaseII pathway needs to be established.

Major points:

1. Figure 1: The authors make the point that about 1/3 of all PGCs die in *Drosophila* and that this process is non-apoptotic. These conclusions have been reached by previous studies (see references above). The authors use additional mediators of apoptotic death to support their point, however, the conclusions do not go beyond previous findings, and it needs to be acknowledged that this is not a new observation.
2. Figure 2 and subsequent figures, such as fig 5d-e) The germ cell survival effects observed could be due to different developmental events: one- as the authors suggest survival of PGCs in the midline, or alternatively, as has previously been shown, PGCs, which fail to migrate or do not transit the midgut, do not undergo cell death and are 'inert' to wunen depletion (Renault et al 2010). Many of the germ cells shown in Figs 2 and 5 appear to be caught in the gut rather than having transited the gut and reached the midline. Thus failure of germ cell death could be due to a migration defect rather than a direct effect on a cell death pathway. To confirm that germ cells are indeed left in the midline, and exited the gut, the authors need to co-localize gut and PGCs and observe earlier stages (stage 10 early 11).
3. Figure 2 previous experiments demonstrated a role for P53 in apoptotic death of germ cells together with the gene outsider (Yamada et al 2008). More recent studies demonstrated that p53 and wunen act in parallel pathways (Slaidina et al 2017). The authors need to show a direct link between P53 and other components of the Parthanatos pathway.
4. The observations made by the authors leave unclear how germ cell death is specifically triggered in the midline. Previous studies showed that the LPP homologs, wunen and wunen2, are expressed in the midline and their function was linked to non-apoptotic death pathway (Starz-Gaiano et al 2001; Renault et al 2004, 2010; Hanyu-Nakamura et al 2004,). In zygotic wunen mutants midline germ cells do not die. Thus Wunen loss and gain of function experiments could be used to link the midline trigger of PGC death to the PARP/AIF pathway. For example, can AIF OE or overexpression of any of the other genes shown in promote germ cell death overcome the zygotic wunen mutant phenotype, or can loss of function of AIF, or DNaseI rescue the death caused by wunen/ or wunen 2 overexpression. Further, depletion of Wunen 2 specifically from germ cells leads to massive (cell autonomous) germ cell death (without a migration defect), can this death be rescued by any of the mutants studied? The authors could further use their DNaseII antibodies/gH2AX to determine relocation to the nucleus upon wunen over-expression. Together, these experiments would determine whether the parthanatos pathway

mediates Wunen induced PGC death.

Minor points:

1. Interestingly and as nicely illustrated by DNaseII mutants, the gene products that trigger germ cell death are provided maternally, thus for zygotically activated RNAi (UAS provided by father, nos-gal4 mostly in germ cells provided by mother), the RNAi is targeting maternally deposited mRNA. The authors need to show that early PGCs contain the respective RNAs tested and that there is indeed a reduction in RNA levels after RNAi. How many RNAi lines were used/per gene?
2. Additional controls should be provided by comparing not just the % of germ cells surviving (in midline or gut) but also the control cross with control UAS transgene.
3. Extended figure 3, *orc2* and *bub3* seem to only have an effect on germ cell survival in overexpression experiments, the observation could thus be non-physiological.
4. Figure 4h, what is the explanation that OE of DNaseII or Aif in germ cells kills germ cells, while PGCs survive when both are overexpressed? The rationale for calling this result 'cooperativity' is rather unclear. A simpler explanation could be competition for gal4.
5. Statistics: Box and whisker blots would provide a more accurate reflection of the data. It is unclear why in some graphs the authors count PGCs and in others a % of embryos with PGCs in 'midline' is given. A more uniform analysis would make comparison between experiments easier.
6. Better designation of overexpression perhaps a "+" after the gene name

Reviewer #2 (Remarks to the Author):

In the paper titled "DNase II triggers a developmental form of cell death by parthanatos", Tarayrah-Ibraheim et al. examined the role of DNase II in *Drosophila* primordial germ cells (PGCs) death. This work is interesting study with use of a variety genetic *Drosophila* models to firstly identify the physiological role of parthanatos in PGCs elimination during early embryogenesis. However, there are some concerns that are not fully developed and leave important questions unanswered.

To measure the dying PGCs, the authors rely on the assay that counts the difference in the number of PGCs between ES10 and 13 throughout the manuscripts. As authors stated, however, some dying PGCs can often still be visualized in the midline region of ES 13 embryos, which makes bias during manual counting. It's important to add the criteria how they counts these "asynchronous cells". Also, it would be better to show together with cell death markers.

The authors nicely used different overexpression or knock down/mutant *drosophila* models to show the selective cell death by DNaseII-dependent parthanatos, which is a key idea in this manuscript. To make convincing conclusion, the author should show the expression of each overexpressed or knockdown proteins in their fly lines (i.e. caspase-dependent cell death proteins, autophagic proteins, lysosomal proteins and parthanatos cell death proteins).

What is the main difference between midline cells in ES10 vs. migrated cells in ES13? Why only midline cells are dying in specific times? Have the author checked spatiotemporal expression of DNase II? What is the fate of remaining PGCs in DNase II knockdown embryos after 13 days? Are they eventually going to dying or longer remaining? It's important question whether DNase II-dependent cell death are temporal or long-lasting events during development.

In Figure 3, lysosomal localization of DNase II is not clear. The authors need to use co-stain with DNase II and lysosomal markers (i.e. LAMP). In addition, nuclear DNase II signal in dying PGCs doesn't look like merging with Hoechst signal, which make data not convincing. Z-stack images and/or

other biochemical evidence should be provided. Nuclear translocation of DNase II is not quantified.

In Figure 4, why did the authors test TUNEL assay in ES 10 and DNase II-cuts assay in ES 13? To show the DNase II-cut-positive but TUNEL-negative cell death, they should test in the same ES days. Also, in Figure 4c, is that right place of PGCs with circle? Why are there less gonadal PGCs and lots of DNase II-cuts signal that are not colocalized with Vasa markers?

There is no direct evidence of AIF-dependent DNase II translocation. Does AIF directly interact with DNase II? Are they co-translocate into nucleus? Also, they only showed knockdown of cyclophilin A-like proteins partially reduced PGC death, but failed to show correlation with AIF and DNase II. Are they interacting with AIF and DNase II? Are they co-translocate into nucleus together with AIF and DNase II?

AIF is associated with and facilitates the nuclear translocation of different nucleases including Endo G, LEI/LDNase II and MIF as the author stated. Among them, MIF has been known for a key regulator of parthanatos (Wang et al., Science, 2016). The author should compare the role of MIF with DNase II in PGC death.

The positive feedback loop is confusing. The first step of parthanatos is DNA damage-induced hyperactivation of PARP-1 and accumulation and translocation of PAR to cytoplasm. The PAR-dependent nuclear translocation of AIF and MIF is critical for the following DNA fragmentation and cell death (Park H et al., Poly (ADP-ribose)(PAR)-dependent cell death in neurodegenerative diseases. International Review of Cell and Molecular Biology, 2020). The authors suggest that DNase II translocate to nucleus to induce DNA damage and PARP-1 activation as the first step and PAR-AIF-DNase II translocate to nucleus might be the second step for cell death. If AIF is required for nuclear translocation of DNase II as they suggested, how can DNase II translocate without help of PAR/AIF in the first step? How these separate translocations of DNase II differentially contribute to PARP-1 activation (in first step) and cell death (in second step)?

Minor concerns;

1. For better understanding to readers in broad fields of biology, please add human homolog of fly genes when they are used in first.
2. In Figure 1, Diap1 data is missing.
3. In Figure 3g, the typical dot pattern of γ -H2Ax signal colocalized with Hoechst is hard to see. Please replace to better representative images with higher magnification.
4. Some quantification are missing as describe above (i.e. Figure 3, Figure 4, Figure 7)
5. The authors nicely tried to address the heterogeneous PAR localization during the process in Figure 7. It would be appreciated if they add the quantification.

Reviewer #3 (Remarks to the Author):

Tarayrah and Arama report an unusual mode of non-apoptotic, regulated cell death (RCD), distinct from GCD, during *Drosophila* development. Using genetic analysis, the authors deduce that the pathway of RCD in PGCs in the fly embryo is likely parthanatos. The authors go on to propose a mechanism for cell death via parthanatos. Although it is thought that DNA cleavage and cell death during parthanatos is dependent on the nuclear translocation of AIF, the mechanism of AIF-induced DNA cleavage is not fully understood. The model presented here is that AIF mediates DNA cleavage

via facilitating the nuclear translocation of the lysosomal DNA endonuclease Dnase II.

The paper addresses an important yet murky area of modern biology: non-apoptotic RCD. The findings of the paper are interesting and potentially important. However, the extent to which Dnase II-dependent DNA cleavage contributes toward PGC cell death is difficult to ascertain from the data presented. A quantitative analysis of the frequencies of PGCs showing nuclear translocation of Dnase II and DNA cleavage, across mutant backgrounds, will provide a more convincing description of what appears to be a novel pathway for non-apoptotic RCD. The overlap in the genes required for pruning PGCs and those required for parthanatos is undoubtedly provocative. What is not clear, however, is that the reduction in the numbers PGCs is entirely due to parthanatos, some variant of this pathway, or a complex program of RCD. Nevertheless, the roles of Dnase II and PARP and their interaction described in this report will undoubtedly be of interest to those working on the parathanatos pathway.

Comments

[1]. The authors hypothesize that the decrease in frequencies of PGCs from ES 10-ES13, a period of 4-5 h, is due to a non-apoptotic mechanism of cell death. They show that mutations in genes involved in such pathways abolish the observed decrease numbers of PGCs. It would be helpful if alternate explanations for the altered numbers of PGCs in these genetic backgrounds are also discussed in the text. Otherwise the narrative appears rather contrived.

For example, is it clear that rates of cell proliferation do not impact the frequencies of PGCs between ES10/ES13?

Along the same lines, is it possible to distinguish between defects in a cell death program and defects in processes such as cell migration that can also influence cell death? If so, the logic requires to be clearly spelled out in the text. If not then this should be mentioned. On Pg. 6, the authors write "It is noteworthy that the ectopically surviving PGCs remained at the midline of the mutant embryos, suggesting that migratory arrest is a distinct and possibly prerequisite process to PGC death.". While the accumulation of PGCs at the midline is clear in dnase II (M, RI) mutants, it is less clear in dor4 mutants and not at all obvious in the car mutants.

[2] The extent to which Dnase II-dependent DNA cleavage contributes toward PGC cell death is difficult to ascertain from the data presented.

In Pg. 8 the authors point out "Moreover, closer examination of the stained ES 10 PGCs revealed that, whereas in the majority of the PGCs DNase II was confined to small cytoplasmic vacuoles (presumably the lysosomes), dying PGCs, revealed by their condensed and distorted morphology, displayed strong nuclear DNase II localization, implying that translocation of DNase II to the nucleus may constitute one of the earliest events of PGC death." It is not clear from the images in Fig. 3 that cells with "condensed and distorted morphology" are the ones with nuclear dnase II. This is important to show. What are the frequencies of cells with "condensed and distorted morphology"? What are the frequencies of cells with nuclear dnase II? Are these frequencies per embryo consistent with a significant role for nuclear dnase II in cell death?

The Top1-mediated ligation assay suggests that PGCs do exhibit dnase II-dependent DNA cleavage. Do the cells that exhibit dnase II cuts have a "condensed and distorted morphology"? Do cells with dnase II cuts co-stain for nuclear dnase II? The data in Fig 4 suggests that the frequency of Top1-mediated ligation events is lower in dnase II mutants. This is an important result and the results of the assays in Fig 4 should be quantified.

[3] The authors find that ATR/Chk1 mutants do not lose PGCs over time. How ATR/Chk1

contribute toward Dnase II dependent cell death is not clear. Cells that escape cell death in Chk1 mutants have nuclear Dnase II suggesting that ATR/Chk1 act downstream to nuclear translocation of Dnase II to initiate cell death. Do the cells with nuclear Dnase II exhibit any Dnase II-cuts or show any evidence for DNA breaks? If so, this would strengthen the connection between nuclear Dnase II and DNA cleavage. These cells should undergo apoptosis at a later stage. If not then the data would suggest that ATR/Chk1 are necessary for the activation of endonuclease activity. This would be important.

[4] The requirement for AIF/PARP and the implication of parthanatos as the cell death program in PGCs.

The aif RI and the PARP RI mutant phenotypes with respect to preventing reduction of PGCs are not completely penetrant. Dnase II staining in aif/PARP mutants that show that Dnase II fails to accumulate in the nucleus. Here again, quantitation of the frequencies of cells expressing nuclear Dnase II would be helpful to assess the contribution of AIF/PARP to the overall process. In other words, is Dnase II accumulation observed in all mutants? It is possible that Dnase II translocation and activation in PGCs involves multiple genes including AIF/PARP.

Although genes like AIF/PARP, with roles in parthanatos are required for the observed reduction in PGC number, whether the involvement of these genes shows that the program of cell death is parthanatos is not clear. It is plausible that the program in PGCs is a variant of this pathway, or a more complex program of cell death.

Arguably, the program of cell death in PGCs is an unusual example of non-apoptotic Dnase II-dependent cell death with features of the parthanatos pathway. While the title of the paper "Dnase II triggers a form of developmental cell death by parthanatos" may be reasonable, it may be an oversimplification.

[5] If any or all of the RNAi lines used in this study have been functionally validated in other studies then these studies should be cited.

[6] The titles of Figure 2 "Inactivation of the lysosomal endonuclease and compromised lysosomal biogenesis and leakage, all block PGC death" and Extended Figure 1 "PGC death is distinct from lysosomal-dependent or autophagy-dependent cell death" appear contradictory.

[7] Extended Figure 1j, please check numbers for nos>atg5,6,8 RI at ES13. The numbers reported and the graph do not tally.

[8] The abbreviation OE requires to be defined.

We would like to express our appreciation of the reviewers' insights and comments which helped us to improve the paper significantly. We essentially addressed all the comments, most of which by performing additional experiments and adding more results.

Note that validations of reagents and gene expression data requested by the reviewers are found at the end of this document (Figures R1 and R2) and are not added to the revised paper. The new figures which were added to the revised paper are mentioned here preceded by the word "new" (e.g. new Fig. 3c).

In the interest of clarity and consistency, when missing, we numbered the reviewers' comments. Our replies are in blue.

REVIEWER COMMENTS

Reviewer #1:

This article explores the mechanisms by which primordial germ cells (PGCs) in *Drosophila*, which that failed to reach the somatic gonad, die. The authors provide an interesting link to a nonapoptotic cell death pathway, called Parthanatos, which is characterized by accumulation of Par and nuclear translocation of AIF. Several previous studies in *Drosophila* (Nakamura, Lehmann, Coffman labs) have demonstrated that PGCs, which do not reach the gonad, die and that this death is non-apoptotic. The regulation of germ cell death during *Drosophila* embryogenesis has been attributed to two pathways, the lipidphosphate phosphatase Wunen, which acts both non-autonomously and germ cell autonomously and a P53 dependent pathway that acts independently of Wunen/Wunen2 (Starz-Gaiano et al 2001; Renault et al 2004, 2010; Hanyu-Nakamura et al 2004, Yamada et al 2008, Slaidina et al 2017). The present manuscript presents new evidence regarding the cellular mechanism of this cell death, however how the parthanatos pathway links to these known pathways, which have been shown to trigger PGC death, needs to be addressed. Furthermore, throughout the manuscript many components are identified that affect PGC death, in some case a physiological role for these genes in PGCs need to be demonstrated, for others a link to the AIF/DNaseII pathway needs to be established.

We thank the reviewer for the constructive comments. We believe that our replies to the reviewers' specific points address the abovementioned concerns.

Major points:

1. Figure 1: The authors make the point that about 1/3 of all PGCs die in *Drosophila* and that this process is non-apoptotic. These conclusions have been reached by previous studies (see references above). The authors use additional mediators of apoptotic death to support their point, however, the conclusions do not go beyond previous findings, and it needs to be acknowledged that this is not a new observation.

As noted by the reviewer, the idea that some of the PGCs undergo developmental cell death between embryonic stages (ES) 10-13 has been previously reported. Furthermore, based on mainly negative results (genetic attempts to inhibit apoptosis in the PGCs), it has also been suggested that this cell death pathway is distinct from classical apoptosis. However, the underlying mechanisms and components involved in this alternative cell death pathway

remained unknown. Accordingly, the aim of the current study was to unravel the pathway and components underlying developmental PGC death, which had been elusive and puzzling for almost two decades. Whereas in the first paragraph of the *Results*, we referred to the previous attempts to inhibit PGC death by genetically manipulating apoptotic proteins, we understand that the previous findings shall be more pronouncedly mentioned throughout the paper. The revised main text now refers to these findings both in the *Abstract* and at the end of the first paragraph of the *Results*, as well as throughout the *Results* and *Discussion* sections when relevant.

2. Figure 2 and subsequent figures, such as fig 5d-e) The germ cell survival effects observed could be due to different developmental events: one- as the authors suggest survival of PGCs in the midline, or alternatively, as has previously been shown, PGCs, which fail to migrate or do not transit the midgut, do not undergo cell death and are 'inert' to wunen depletion (Renault et al 2010). Many of the germ cells shown in Figs 2 and 5 appear to be caught in the gut rather than having transited the gut and reached the midline. Thus failure of germ cell death could be due to a migration defect rather than a direct effect on a cell death pathway. To confirm that germ cells are indeed left in the midline, and exited the gut, the authors need to co-localize gut and PGCs and observe earlier stages (stage 10 early 11).

We performed the experiment suggested by the reviewer and directly visualized the PGCs and the midgut in the *dnasell* mutants at ES 10, 11, 12, and 13. The results, which are presented in new Supplementary Fig. 2, demonstrate that both the gonadal PGCs and the ectopically surviving midline PGCs transverse the midgut in the *dnasell* mutant embryos, indicating that DNase II is not required for PGC migration. Consistent with this idea, when we examined embryos maternally mutant for both the *wunens* and *dnasell*, the surviving PGCs (both the gonadal and the midline subsets) migrated to the gonads (see also the relevant paragraph in our reply to comment #3 of this reviewer).

3. Figure 2 previous experiments demonstrated a role for P53 in apoptotic death of germ cells together with the gene outsider (Yamada et al 2008). More recent studies demonstrated that p53 and wunen act in parallel pathways (Slaidina et al 2017). The authors need to show a direct link between P53 and other components of the Parthanatos pathway.

We believe that the reviewer refers to "non-apoptotic" in the sentence "a role for p53 in apoptotic death of germ cells".

The reviewer mentions several previous studies focusing on essentially four components shown to affect PGC survival and death, i.e. p53, Outsiders (Out), and Wunens (for simplicity we refer to the redundant enzymes Wun2 and Wun as Wunens), although the mechanisms by which they mediate cell survival and death remained unclear. Whereas each of these components probably deserves several dedicated studies aimed to uncover their mechanisms of action in PGC death, unraveling the PGC death pathway in the current study shall evidently allow for more directed studies in the framework of parthanatos.

Nonetheless, based on initial experiments (most of which are now included in the revised paper), we provide (in the *Discussion*) new directions to explore the mechanisms by which these components mediate PGC death, and how these components might fit in the parthanatos model:

p53 - The discovery that p53 is involved in PGC death was originally puzzling, as at that time, the prevailing dogma was that p53 mediates apoptotic cell death in *Drosophila*. In the current study, we put forward the idea that p53 affects PGC death through mediating lysosomal membrane permeabilization (LMP), as was previously reported in other systems (Aits and Jaattela, 2013). Several findings in the current paper support this idea. Loss of *p53*, as well as four additional genetic manipulations that attenuate LMP, all phenocopy *dnasell* mutants, inhibiting developmental PGC death cell autonomously, but not normal migration of the gonadal PGC subset (Supplementary Fig. 3h,j; Supplementary Fig. 4b-e). In addition, we now show that in the *p53* mutants, as well as in the other LMP associated mutants, DNase II does not translocate to the nucleus in the ectopically surviving midline PGCs, while more than 75% of the midline dying PGCs in wild-type (WT) embryos displayed nuclear DNase II (new Supplementary Fig. 3k,l; new Supplementary Fig. 4g-i). Furthermore, we now show that overexpression (OE) of AIF restores PGC death in *p53* knockdown embryos, but does not cause excessive PGC death in control embryos, demonstrating that *p53* genetically interacts with *aif* during PGC death (new Supplementary Fig. 3j,m).

Out - Mutations in the *out* gene, which codes for a monocarboxylate transporter (MCT), were reported to attenuate PGC death, and it was suggested that Out and p53 may operate in a common cell death pathway, with p53 functioning downstream of Out (Yamada et al., 2008). How an MCT might affect cell death remains unclear. We hypothesized that since PARP-1 generates PAR polymers by catalyzing the polymerization of ADP-ribose units from donor nicotinamide adenine dinucleotide (NAD⁺) molecules, mutations in the *out* gene might attenuate PGC death by significantly decreasing the cellular pool of NAD⁺. The logic behind this hypothesis is based on a recent report, showing that inhibition of two MCTs in cancer cells led to accumulation of high intracellular lactate, which in turn inhibited lactate dehydrogenase, one of two enzymes regenerating NAD⁺ from NADH (Benjamin et al., 2018). We tested this possibility, by genetically manipulating the cellular pool of NAD⁺, and the findings are consistent with this hypothesis, but the data is too preliminary to be included in the paper.

Wunens - The roles of Wunen (Wun) and Wun2, two Lipid Phosphate Phosphatase (LPP) enzymes which act redundantly in the germ cells and the soma, to regulate PGC migration and death were extensively studied (Renault and Lehmann, 2006). However, there are several major differences between p53 and the Wunens during PGC death; Whereas inactivation of p53 and Out attenuates developmental PGC death, inactivation of Wunens in the PGCs was shown to trigger excessive non-apoptotic cell death. Therefore, in contrast to DNase II and the other components in the PGC death pathway (including p53 and Out), which act at the level of the cell death machinery to promote PGC death (and not PGC migration), the Wunens act as survival factors, presumably acting upstream to regulate PGC migration and death. Importantly, as suggested in the next point of this reviewer, we now generated embryos mutant for both the *wunens* and *dnasell*, showing that DNase II is also involved in the Wunens signaling-induced PGC death.

4. The observations made by the authors leave unclear how germ cell death is specifically triggered in the midline. Previous studies showed that the LPP homologs, *wunen* and *wunen2*, are expressed in the midline and their function was linked to non-apoptotic death pathway (Starz-Gaiano et al 2001; Renault et al 2004, 2010; Hanyu-Nakamura et al 2004,). In zygotic *wunen* mutants midline germ cells do not die. Thus *Wunen* loss and gain of function experiments could be used to link the midline trigger of PGC death to the PARP/AIF pathway. For example, can AIF OE or overexpression of any of the other genes shown in promote germ cell death overcome the zygotic *wunen* mutant phenotype, or can loss of function of AIF, or DNaseI rescue the death caused by *wunen*/ or *wunen 2* overexpression. Further, depletion of *Wunen 2* specifically from germ cells leads to massive (cell autonomous) germ cell death (without a migration defect), can this death be rescued by any of the mutants studied? The authors could further use their DNaseII antibodies/gH2AX to determine relocation to the nucleus upon *wunen* over-expression. Together, these experiments would determine whether the parthanatos pathway mediates *Wunen* induced PGC death.

We now performed genetic experiments to test whether PGC death induced by the lack of maternal *wunens* might also be mediated by the DNase II pathway. The experiments that are presented in new Supplementary Fig. 3a-g, took advantage of a strong *wunens* mutant allele, *wun2^{N14}*, which has been reported to inactivate both *wun2* and *wun* (presumably coding for a dominant-negative form of *Wun2* that also inhibits *Wun*), causing excessive PGC death in embryos from mothers carrying this allele over a deficiency (Hanyu-Nakamura et al., 2004; Renault et al., 2010).

The details of the results are found in the main text. Concisely, embryos from mothers double mutant for both *dnaseII* and *wunens* displayed partial rescue of PGC death as compared with embryos from *wunens* only mutant mothers, linking *Wunens* survival signaling with DNase II-mediated PGC death pathway, placing the latter downstream of *Wunens* the (please refer to the relevant part in the *Results* for details). Furthermore, this experiment also uncoupled between PGC death and migration in the *wunens* mutant background, highlighting an additional important point; in a few double mutants all the PGCs were rescued and almost all of them migrated to the gonads, demonstrating that in the absence of *wunens*, the midline and gonadal subsets of PGCs have similar capability to migrate to the gonads. This is also consistent with (Slaidina and Lehmann, 2017), who suggested that the levels of *wunens* in the germlasm taken by the PGCs during early embryonic stages, pre-determine which of the PGCs would remain in the midline during ES 10-13 and undergo cell death.

Minor points:

1. Interestingly and as nicely illustrated by DNaseII mutants, the gene products that trigger germ cell death are provided maternally, thus for zygotically activated RNAi (UAS provided by father, nos-gal4 mostly in germ cells provided by mother), the RNAi is targeting maternally deposited mRNA. The authors need to show that early PGCs contain the respective RNAs tested and that there is indeed a reduction in RNA levels after RNAi. How many RNAi lines were used/per gene?

In this study we used 26 RNAi lines against 25 genes, 10 OE lines of 10 genes, and 2 dominant negative (DN) lines. Fly lines for which specificity and efficiency were previously validated in other studies, including previous studies from our lab, are now all indicated with references to the original studies in the *Methods* section under the “*Drosophila strains and crosses*” subtitle.

We also validated OE and knockdown fly lines which were used for the first time in the current study (Fig. R1; a reviewer only figure at the end of this document). Knockdown validations of *dnasell* and *aif* were performed using antibodies; for *dnasell^{Ri}*, anti-DNase II antibodies generated in the current study were used in immunostaining; for *aif^{Ri}*, anti-AIF antibodies generated in the current study and only work in Western blotting were used for validation. To examine the efficiencies and specificities of the other non-validated RNAi (and OE) lines, we performed real-time PCR on RNAs extracted from ES 1-2 embryos laid by mothers expressing the early maternal driver *mat- α -tubulin-Gal4-VP16* (abbreviated *mata*) and the appropriate *UAS-RNAi* or *UAS-OE* transgene. In these mothers, the corresponding RNAs were already knocked down or overexpressed during oogenesis.

Note, in addition to these validations, when possible (depending on the availability of non-lethal alleles), we also used maternal mutant lines which were previously validated, including mutants for the major PGC death mediators, DNase II and AIF (*dnasell^o* and *aif^{f52/T2}*).

To validate that early PGCs contain the respective RNAs, we turned to two databases with *in situ* RNA hybridizations in *Drosophila* embryos, called the Berkeley *Drosophila* Genome Project (BDGP) *in situ* database and Fly-FISH (Fig. R2; a reviewer only figure at the end of this document). Each database contains a different set of genes and embryonic stages. Importantly, these databases demonstrate general (ES 1-3) and PGC-specific (ES 4-8) expression of maternal RNAs for the major mediators of PGC death, DNase II, AIF, Parp1, and Cyp1. Three additional lysosomal genes, *car*, *cathB* and *cathL*, as well as the DDR checkpoint gene *chk1*, were also expressed maternally and in the early PGCs. Note that genes that do not appear in this figure are also missing in the databases.

2. Additional controls should be provided by comparing not just the % of germ cells surviving (in midline or gut) but also the control cross with control UAS transgene.

These controls are now added in Fig. 1i (*nos-Gal4*) and Fig. 1j (*nos-Gal4>eGFP^{Ri}*).

3. Extended figure 3, *orc2* and *bub3* seem to only have an effect on germ cell survival in overexpression experiments, the observation could thus be non-physiological.

OE of *orc2* and *bub3* (now appearing in Supplementary Fig. 7) were used to demonstrate the involvement in PGC death of other checkpoints suggested to interact with the DDR. Although OE is non-physiological, these are still highly informative, showing that manipulating the DDR associated checkpoints attenuate PGC death downstream of DNase II nuclear translocation. Furthermore, in Supplementary Fig. 7d, we demonstrate genetic (physiological) interaction between *orc2* and *dnasell*, showing that the block in PGC death in the *dnasell* mutants is completely bypassed when *orc2* is inactivated, while *orc2* knockdown alone does not affect the normal rate of PGC death. The finding that nuclear DNase II could be readily detected in

ectopically surviving midline PGCs overexpressing *bub3* indicates that the DDR associated checkpoints are further downstream in this pathway (Supplementary Fig. 7h-j).

4. Figure 4h, what is the explanation that OE of DNaseII or Aif in germ cells kills germ cells, while PGCs survive when both are overexpressed? The rationale for calling this result 'cooperativity' is rather unclear. A simpler explanation could be competition for gal4.

In this comment, the reviewer likely referred to Supplementary Fig. 4h (which is now Supplementary Fig. 8h). The data presented in this graph was inversely interpreted by the reviewer, presumably because the previous version of the paper utilized column charts to present both ectopically surviving PGCs and excessive PGC death. The paper now utilizes box and whiskers charts to present the ectopic survival of midline PGCs, and column charts to present excessive PGC death by quantifying the number of embryos containing less than 15 or 10 PGCs at ES 13, as the average number of PGCs in control embryos at that stage is 19.

5. Statistics: Box and whisker blots would provide a more accurate reflection of the data. It is unclear why in some graphs the authors count PGCs and in others a % of embryos with PGCs in 'midline' is given. A more uniform analysis would make comparison between experiments easier. We thank the reviewer for this suggestion.

6. Better designation of overexpression perhaps a "+" after the gene name We now added OE in superscript to emphasize overexpression.

Reviewer #2:

In the paper titled "DNase II triggers a developmental form of cell death by parthanatos", Tarayrah-Ibraheim et al. examined the role of DNase II in *Drosophila* primordial germ cells (PGCs) death. This work is interesting study with use of a variety genetic *Drosophila* models to firstly identify the physiological role of parthanatos in PGCs elimination during early embryogenesis. However, there are some concerns that are not fully developed and leave important questions unanswered.

We thank the reviewer for the constructive comments. We believe that our replies to the reviewers' specific points address the abovementioned concerns.

1. To measure the dying PGCs, the authors rely on the assay that counts the difference in the number of PGCs between ES10 and 13 throughout the manuscripts. As authors stated, however, some dying PGCs can often still be visualized in the midline region of ES 13 embryos, which makes bias during manual counting. It's important to add the criteria how they counts these "asynchronous cells". Also, it would be better to show together with cell death markers.

Manually counting and calculating the difference in PGC numbers between ES 10 and 13/14 constitutes a gold standard in this field and is well established as an accurate determination of the number of dying PGCs; to the best of our knowledge, this method has been utilized in essentially all the PGC death studies during the past two decades (Hanyu-Nakamura et al., 2004; Renault et al., 2004, 2010; Sano et al., 2005; Yamada et al., 2008; Slaidina and Lehmann, 2017).

The accuracy of this method stems from the fact that the total number of PGCs, around 35, is manageable for manual counting under the confocal microscope following staining with anti-Vasa antibodies. Because the PGCs die in an asynchronous manner, highly condensed and distorted PGCs with rather fainter Vasa staining signal (i.e. PGCs in advanced cell demolition stages) are readily detected throughout ES 10-13, and are evidently distinct from the living PGCs. At ES 13, most of the midline PGC subset is already eliminated, and the few dying PGCs that are left, are thus easily detected and omitted from our counts.

The exact counting criteria is updated accordingly and can be found in the *Methods* section under the subtitle "*Quantification of PGC death*".

2. The authors nicely used different overexpression or knock down/mutant drosophila models to show the selective cell death by DNaseII-dependent parthanatos, which is a key idea in this manuscript. To make convincing conclusion, the author should show the expression of each overexpressed or knockdown proteins in their fly lines (i.e. caspase-dependent cell death proteins, autophagic proteins, lysosomal proteins and parthanatos cell death proteins).

Please see our reply to a similar point raised by reviewer #1 (Minor point #1).

3. What is the main difference between midline cells in ES10 vs. migrated cells in ES13? Why only midline cells are dying in specific times?

At ES 10, all the PGCs which were carried with the midgut primordium, transverse the midgut and appear near the midline area. They then divide into two major subsets, which we dub the midline subset (encompassing one-third of the total number of PGCs) and the gonadal subset (encompassing two-thirds of the total number of PGCs). The subset of gonadal PGCs further sorts bilaterally and migrates toward the somatic precursors of the two gonads (each gonad at ES 13 eventually consists of about one-third of the total number of PGCs at ES 10), while the midline subset remains at around the midline area and is eliminated by cell death (these are the developmentally dying PGCs).

The reason why one-third of the PGCs are dying has been attributed to inheritance of different levels of maternal factors during early stages of PGC specification. Specifically, it was shown that the maternally inherited levels of the two Lipid Phosphate Phosphatases, Wun and Wun2, pre-determine the two PGC subsets, the midline subset that would undergo cell death and the living gonadal subset (Slaidina and Lehmann, 2017). The revised paper includes more experimental data and hypotheses concerning a possible link between the Wunens and the parthanatos machinery of PGC death.

4. Have the author checked spatiotemporal expression of DNase II?

Our genetic data demonstrate that maternal DNase II is required for PGC death. This is now also supported by *in situ* RNA hybridizations from a public database, showing that *dnaseII* RNA is present at early embryonic stages in the syncytial embryo and in the pole/PGC cells (a stage in which the PGCs are formed; Fig. R2, a reviewer only figure at the end of this document).

Moreover, using our validated anti-DNase II antibodies, we followed spatial expression of DNase II in the living and dying PGCs. We show that at ES 10, DNase II is highly expressed in all the PGCs, such that these cells are standing out by virtue of their high DNase II expression (Fig. 3a,b). In the living PGCs, DNase II is confined to the lysosomes, as indicated by co-localization with a lysosomal marker (new Fig. 3c), whereas in the dying PGCs, DNase II is released from the lysosomes (new Fig. 3d) and translocates to the nucleus (new Fig. 3e,f). Finally, a high resolution movie constructed from multiple Z stacks demonstrates the non-nuclear cytoplasmic confinement of DNase II in a living PGC versus the nuclear localization of DNase II in a dying PGC (new Movie 1).

5. What is the fate of remaining PGCs in DNase II knockdown embryos after 13 days? Are they eventually going to dying or longer remaining? It's important question whether DNase II-dependent cell death are temporal or long-lasting events during development.

We monitored the ectopically surviving PGCs in the *dnaseII* mutants to until the end of embryonic development when the larval structures are formed (during ES 14-17). As shown in new Fig. 2f-j, these PGCs remained intact and were positioned ectopic to the gonads throughout embryogenesis, indicating that DNase II involvement in PGC death is long-lasting and essential for the elimination of these cells.

6. In Figure 3, lysosomal localization of DNase II is not clear. The authors need to use co-stain with DNase II and lysosomal markers (i.e. LAMP).

New Fig. 3c,d shows co-staining for DNase II and the lysosomal membrane protein Arl8, demonstrating that in living PGCs, DNase II is localized within lysosomes, and that it is released from the lysosomes in the dying PGCs.

7. In addition, nuclear DNase II signal in dying PGCs doesn't look like merging with Hoechst signal, which make data not convincing. Z-stack images and/or other biochemical evidence should be provided.

In the new Movie 1, we present 3D reconstitution of high resolution Z stack images, demonstrating the nuclear localization of DNase II in dying PGCs. Note that DNase II accumulation in the nucleus is associated with severe DNA fragmentation detected by the elimination of the Hoechst staining signal from the domains where DNase II localization is highly prominent.

8. Nuclear translocation of DNase II is not quantified.

In new Supplementary Fig. 4i, we added quantification of the percentage of midline dying PGCs with nuclear DNase II, showing that 70% of the condensed dying PGCs displayed nuclear DNase II. We also quantified nuclear DNase II translocation in the LMP mutant PGCs (new Supplementary Fig. 4i), *aif* knockdown (new Fig. 6h) and *cyp1* knockdown (new Supplementary Fig. 9e), showing that essentially none of the ectopically surviving midline PGCs in the affected mutants display nuclear DNase II.

9. In Figure 4, why did the authors test TUNEL assay in ES 10 and DNase II-cuts assay in ES 13? To show the DNase II-cut-positive but TUNEL-negative cell death, they should test in the same ES days.

Since PGC death occurs asynchronously during ES 10-13 with a slight peak at ES 11 (see also new Fig. 1d), the dying PGCs can be identified at any time during this period, with the most chances to detect them at ES 11. Therefore, the exact embryonic stage is meaningless, as long as we can detect a dying PGC (highly condensed and distorted morphology with faint Vasa signal).

In contrast, for PGC-specific OE of the pro-apoptotic gene *hid*, which was used as a negative control in the DNase II-cuts assay (Fig. 4g), and as a positive control for TUNEL labeling of apoptotic PGCs (Supplementary Fig. 6b), the setup is different. The reason for the difference is that the zygotic expression starts at ES 9 (hence also the expression of *Hid*), so we must adhere to ES 10 embryos for detection of the apoptotic PGCs, since the majority of the PGCs in that background are already eliminated by ES 13.

10. Also, in Figure 4c, is that right place of PGCs with circle? Why are there less gonadal PGCs and lots of DNase II-cuts signal that are not colocalized with Vasa markers?

The circled PGC (now shown in Fig. 4e) is indeed a dying midline PGC. The reason why this might have been confusing is because the embryos is visualized in a side view. However, we clearly recognize that this is the correct place for midline PGCs, as it is located immediately above the outline of the CNS (background Vasa staining). This is also the reason why we do not see all the gonadal PGCs, as in this view of the embryo not all of the PGCs are in the same focal plane.

DNase II-cuts in cells other than PGCs presumably reflect developmental cell death of different cell types in the embryo. DNase II is known to function in phagocytes to further cleave the DNA of engulfed apoptotic cells.

It is important to note that this assay is not trivial for detection of the dying PGCs, both because of the asynchronous nature of developmental PGC death and even more so, because of the transient nature of this signal in the dying PGCs, which appears after sufficient accumulation of DNA breaks, and then disappears once the DNA is highly fragmented.

11. There is no direct evidence of AIF-dependent DNase II translocation.

Fig. 6e-g and the quantification in new Fig. 6h, show that about 80% of the ectopically surviving midline PGCs in the *aif* knockdown embryos displayed non-nuclear DNase II. In contrast, in the control embryos, only 30% of the dying midline PGCs exhibited non-nuclear DNase II (new Supplementary Fig. 4i).

12. Does AIF direct interact to DNase II?

New Supplementary Fig. 9a presents co-immunoprecipitation assays demonstrating that recombinant full-length AIF can physically associate with DNase II *in vitro*.

13. Are they co-translocate into nucleus?

Unfortunately, at this point, we cannot perform this experiment, as the antibodies that we generated against *Drosophila* AIF, and which nicely worked in Western blotting (Fig. R1i at the end of this document), do not work in immunostaining.

However, nuclear translocation of mammalian AIF during non-apoptotic cell death has been previously demonstrated in several studies, including in parthanatos models of cell death (Wang et al., 2016). Moreover, in the current study, we present compelling evidence for the involvement of AIF in DNase II-mediated PGC death:

1. AIF OE restored normal PGC death levels in *dnaseII* mutant embryos (Fig. 6c,d).
2. Double OE of DNase II and AIF (but not single expression of each of them) induced precocious non-apoptotic PGC death (Supplementary Fig. 8b-h).
3. DNase II nuclear translocation in midline PGCs is severely attenuated upon *aif* knockdown (new Fig. 6e-h).
4. AIF binds to DNase II *in vitro* (new Supplementary Fig. 9a).
5. CypA, which has been previously shown to bind to AIF and mediate its nuclear translocation and associated nuclease activity in mammalian systems (Candé et al., 2004b, 2004a), physically associates with *Drosophila* AIF *in vitro*, and is required for DNase II nuclear translocation and PGC death (new Supplementary Fig. 9b-g).

14. Also, they only showed knockdown of cyclophilin A-like proteins partially reduced PGC death, but failed to show correlation with AIF and DNase II. Are they interacting with AIF and DNase II? Are they co-translocate into nucleus together with AIF and DNase II?

The link between CypA and the AIF-DNase II mediated PGC death is now significantly expanded. New Supplementary Fig. 9g presents pulldown assays demonstrating that recombinant full-length AIF can physically associate with Cyp1, a *Drosophila* CypA ortholog, *in vitro*. In addition, new Supplementary Fig. 9d,e shows that 100% of the ectopically surviving midline PGCs in the *cyp1* knockdown embryos displayed non-nuclear DNase II (as opposed to only 30% in the control embryos), indicating that Cyp1 affects DNase II nuclear translocation. Finally, new Supplementary Fig. 9c presents genetic interaction between *aif* and *cyp1*, showing that OE of *aif* restored PGC death in *cyp1* knockdown embryos (while OE of *aif* in an otherwise wild-type background did not affect PGC death levels; Fig. 6b).

15. AIF is associated with and facilitates the nuclear translocation of different nucleases including Endo G, LEI/LDNase II and MIF as the author stated. Among them, MIF has been known for a key regulator of parthanatos (Wang et al., Science, 2016). The author should compare the role of MIF with DNase II in PGC death.

AIF has been suggested to interact with several distinct nucleases in different model systems, the context of which has been always non-apoptotic cell death, although not all studies defined the exact cell death pathway as parthanatos (either because of historical bias or lack of additional data). The *Drosophila* genome does not contain gene orthologs for *MIF* or *LEI*. Furthermore, we detected no effect on PGC death in a well characterized *endoG* loss-of-function mutant (Supplementary Fig. 1e). We now added a paragraph in the *Discussion* section in which we compare between the different parthanatos subtypes and suggest a unified model for defining

cell death by parthanatos. Concisely, whereas PARP-1 and AIF are invariably required for triggering and mediating parthanatos, the identity of the PARP-1-dependent AIF-associated nuclease (PAAN) could vary between different systems and cell types.

16. The positive feedback loop is confusing. The first step of parthanatos is DNA damage-induced hyper-activation of PARP-1 and accumulation and translocation of PAR to cytoplasm. The PAR-dependent nuclear translocation of AIF and MIF is critical for the following DNA fragmentation and cell death (Park H et al., Poly (ADP-ribose)(PAR)-dependent cell death in neurodegenerative diseases. *International Review of Cell and Molecular Biology*, 2020). The authors suggest that DNase II translocate to nucleus to induced DNA damage and PARP-1 activation as the first step and PAR-AIF-DNase II translocate to nucleus might be the second step for cell death. If AIF is required for nuclear translocation of DNase II as they suggested, how can DNase II translocate without help of PAR/AIF in the first step? How these separate translocations of DNase II differentially contribute to PARP-1 activation (in first step) and cell death (in second step)?

The positive feedback amplification loop model is based on our genetic data presented in Supplementary Fig. 10c-i, showing that in the *dnaseII* mutants, the ectopically surviving midline PGCs displayed persistent nuclear PAR signal, whereas in *parp-1* knockdown, the translocation of DNase II to the nucleus in the ectopically surviving midline PGCs was dramatically reduced. Furthermore, data presented in new Supplementary Fig. 5e, suggests that Chk1 activation and the DDR are also important for DNase II nuclear translocation. Whereas this genetic data demonstrates the interdependence between the DNase II/AIF arm and the PARP-1/PAR arm of PGC death (hence the positive feedback amplification loop), it leaves unknown the initial signal and steps that trigger this loop.

One possibility, albeit not exclusive, is that CathB and L, released from the lysosomes, may promote the initial release of AIF from the mitochondria, as was previously reported (Yuste et al., 2005). Consistent with this idea, in Supplementary Fig. 1g-i, we show that knockdowns of *cathB* and *cathL* partially attenuate PGC death. Another possibility, is that the Bcl-2 family members in *Drosophila* could be involved in LMP and/or AIF release, as was proposed for BAX, which was shown to mediate lipid-induced LMP in a caspase-independent manner (Feldstein et al., 2006). Finally, we now demonstrate a link between the survival signal provided by Wunens and DNase II-mediated PGC death by parthanatos (new Supplementary Fig. 3a-g), suggesting that the Wunens signaling might provide initial trigger for PGC death (also refer to the relevant parts in the *Results* and *Discussion* sections). We also updated the model in Fig. 8 to reflect the involvement of the DDR in this loop.

Finally, in new Supplementary Fig. 6c-k, we demonstrate that induction of mild DNA damage by transient OE of an exogenous nuclease is sufficient to trigger non-apoptotic cell death in the PGCs.

Minor concerns;

1. For better understanding to readers in broad fields of biology, please add human homolog of fly genes when they are used in first.

We added the mammalian homologs of all the genes with different fly names.

2. In Figure 1, Diap1 data is missing.

It is included in Fig. 1i

3. In Figure 3g, the typical dot pattern of γ -H2Ax signal colocalized with Hoechst is hard to see. Please replace to better representative images with higher magnification.

We replaced the previous images with higher resolution images (Fig. 4b), but the expression pattern is still not dotted. This could be attributed to the facts that DNA fragmentation in PGC death is much more pronounced than in cellular systems commonly used to investigate DNA damage, and that histone H2Av is widely distributed in the *Drosophila* genome (Leach et al., 2000).

4. Some quantifications are missing as described above (i.e. Figure 3, Figure 4, Figure 7)

Quantifications were added as follows:

The percentage of PGCs with nuclear DNase II in control and several mutants (new Fig. 6h; new Supplementary Fig. 4i; new Supplementary Fig. 9e).

The percentage of embryos with midline PGCs positive for γ H2Av staining in control and *dnasell* mutants (Fig. 4d).

As for the Top I-mediated ligation, we find that although specific when it works, the robustness of this method for detection of DNase II-specific cuts is not optimal for quantitative detection in the dying PGCs, presumably due to the asynchronous nature of PGC death and the transient nature of this signal in the dying PGCs, which seems to appear after sufficient accumulation of DNA breaks and to disappear once the DNA is highly fragmented. Therefore, under these conditions, this method is rather qualitative than quantitative, and its strength in our system stems from the contrast between the positive labeling (albeit scarce) of the developmentally dying PGCs and the negative labeling of the apoptotic PGCs following Hid OE.

5. The authors nicely tried to address the heterogeneous PAR localization during the process in Figure 7. It would be appreciated if they add the quantification.

Quantification of the percentage of PGCs devoid of nuclear PAR is now added in new Fig. 7d.

Reviewer #3:

Tarayrah and Arama report an unusual mode of non-apoptotic, regulated cell death (RCD), distinct from GCD, during *Drosophila* development. Using genetic analysis, the authors deduce that the pathway of RCD in PGCs in the fly embryo is likely parthanatos. The authors go on to propose a mechanism for cell death via parthanatos. Although it is thought that DNA cleavage and cell death during parthanatos is dependent on the nuclear translocation of AIF, the mechanism of AIF-induced DNA cleavage is not fully understood. The model presented here is that AIF mediates DNA cleavage via facilitating the nuclear translocation of the lysosomal DNA endonuclease Dnase II.

The paper addresses an important yet murky area of modern biology: non-apoptotic RCD. The findings of the paper are interesting and potentially important. However, the extent to which Dnase II-dependent DNA cleavage contributes toward PGC cell death is difficult to ascertain from the data presented. A quantitative analysis of the frequencies of PGCs showing nuclear translocation of Dnase II and DNA cleavage, across mutant backgrounds, will provide a more convincing description of what appears to be a novel pathway for non-apoptotic RCD.

We now added quantifications of almost all the assays performed in this paper. Most relevant, we quantified nuclear translocation of DNase II in control and key mutant embryos (new Fig. 6h; new Supplementary Fig. 4i; new Supplementary Fig. 9e), midline PGCs positive for γ H2Av staining in control and *dnaseII* mutants (Fig. 4d), and quantification of the percentage of PGCs devoid of nuclear PAR (new Fig. 7d).

The overlap in the genes required for pruning PGCs and those required for parthanatos is undoubtedly provocative. What is not clear, however, is that the reduction in the numbers PGCs is entirely due to parthanatos, some variant of this pathway, or a complex program of RCD. Nevertheless, the roles of Dnase II and PARP and their interaction described in this report will undoubtedly be of interest to those working on the parathanatos pathway.

In the current study, we show that the key components that molecularly define parthanatos are also required for PGC death, including PARP-1, AIF and the PARP-1-dependent AIF-associated nuclease (PAAN), which in the case of PGC death is DNase II. We also show that a fourth component, cyclophilin A, which in other studies of non-apoptotic cell death has been implicated in AIF nuclear translocation and associated nuclease activity, is also required for PGC death. Furthermore, we demonstrate physical interactions and interdependency of these components and activities (e.g. PAR release from the nucleus and DNase II translocation to the nucleus), in a manner reminiscent of the general parthanatos model. Altogether, our findings are consistent with the idea that PGC death pathway is a developmental subtype of parthanatos. We added a paragraph in the *Discussion* section in which we compare between the different parthanatos subtypes and suggest a unified model for defining cell death by parthanatos. Concisely, whereas PARP-1 and AIF are invariably required for triggering and mediating parthanatos, the identity of the PAAN could vary between different systems and cell types.

Comments

[1]. The authors hypothesize that the decrease in frequencies of PGCs from ES 10-ES13, a period of 4-5 h, is due to a non-apoptotic mechanism of cell death. They show that mutations in genes

involved in such pathways abolish the observed decrease numbers of PGCs. It would be helpful if alternate explanations for the altered numbers of PGCs in these genetic backgrounds are also discussed in the text. Otherwise the narrative appears rather contrived.

For example, is it clear that rates of cell proliferation do not impact the frequencies of PGCs between ES10/ES13?

The idea that the decrease in PGC numbers between ES 10 and 13 is due to cell death has been documented and investigated in several previous studies by different groups (Sonnenblick, 1950; Underwood et al., 1980; Technau and Campos-Ortega, 1986; Hanyu-Nakamura et al., 2004; Renault et al., 2004, 2010; Sano et al., 2005; Yamada et al., 2008; Slaidina and Lehmann, 2017). Furthermore, it was shown that PGCs do not proliferate or transdifferentiate during these embryonic stages (Sonnenblick, 1941, 1950; Underwood et al., 1980; Technau and Campos-Ortega, 1986; Deshpande et al., 1999). Importantly, we and others can clearly detect these condensed and distorted dying PGCs near the embryo midline (for instance in Fig. 1b), and we now clearly show massive fragmentation of the DNA, which is an ultimate characteristic of cell death (new Fig. 3e; new Movie 1; Fig. 4a,b,e).

Along the same lines, is it possible to distinguish between defects in a cell death program and defects in processes such as cell migration that can also influence cell death? If so, the logic requires to be clearly spelled out in the text. If not then this should be mentioned. On Pg. 6, the authors write “It is noteworthy that the ectopically surviving PGCs remained at the midline of the mutant embryos, suggesting that migratory arrest is a distinct and possibly prerequisite process to PGC death.”.

This is indeed an important point which we now unequivocally addressed showing that the DNase II-mediated pathway is specific for PGC death and is not affecting PGC migration (please refer to our reply to a similar point raised by reviewer #1 point 2). The relevant data demonstrating this point are presented in new Supplementary Fig. 2 and new Supplementary Fig. 3d,f,g.

While the accumulation of PGCs at the midline is clear in dnase II (M, RI) mutants, it is less clear in *dor4* mutants and not at all obvious in the *car* mutants.

The partial penetrance of the phenotype in *dor* and *car* is due to the hypomorphic nature of the mutant alleles that we examined, as both of them are essential genes. To increase the phenotypic penetrance, we repeated these experiments by specifically knocking down *dor* and *car* in the PGCs. Previous mutant images and quantifications were removed and replaced with the knockdown counterparts: The images are presented in new Supplementary Fig. 1a,b and the corresponding quantifications in new Fig. 2k.

[2] The extent to which Dnase II-dependent DNA cleavage contributes toward PGC cell death is difficult to ascertain from the data presented.

In Pg. 8 the authors point out “Moreover, closer examination of the stained ES 10 PGCs revealed that, whereas in the majority of the PGCs DNase II was confined to small cytoplasmic vacuoles (presumably the lysosomes), dying PGCs, revealed by their condensed and distorted morphology, displayed strong nuclear DNase II localization, implying that translocation of DNase II to the

nucleus may constitute one of the earliest events of PGC death.” It is not clear from the images in Fig. 3 that cells with “condensed and distorted morphology” are the ones with nuclear dnase II. This is important to show. What are the frequencies of cells with “condensed and distorted morphology”? What are the frequencies of cells with nuclear dnase II? Are these frequencies per embryo consistent with a significant role for nuclear dnase II in cell death?

The detection of the dying PGCs during ES 10-13 is straightforward, as this subset of PGCs does not sort bilaterally and migrate to the gonads, and instead these PGCs remain near the embryo midline during these stages. PGC death during these embryonic stages occurs in an asynchronous manner, meaning that not all the PGCs in this subset die at the same rate, although by ES 13, most of them are already eliminated or found at an advanced cell demolition stages. In new Fig. 1d, we show cell death index in ES 10, 11, 12, and 13, demonstrating the asynchronous nature of PGC death and indicating that the chances to detect dying PGCs at advanced demolition stages are highest during ES 10-12 with a peak at ES 11.

The dying midline PGCs are clearly distinct in their examined molecular features as compared with the gonadal subset of PGCs. In particular, we now quantified DNase II nuclear translocation and PAR release from the nucleus in the midline and gonadal subsets and in several relevant mutants (new Fig. 6h; new Supplementary Fig. 4i; new Supplementary Fig. 9e) and (new Fig. 7d), respectively. These quantifications demonstrate a direct correlation between PGC death and DNase II nuclear translocation and PAR release from the nucleus.

In new Movie 1 (and new Fig. 3f), we present a high resolution movie constructed from multiple Z stacks demonstrating the non-nuclear cytoplasmic confinement of DNase II in a living PGC, versus the nuclear localization of DNase II in a dying PGC. Furthermore, New Fig. 3c,d shows co-staining for DNase II and the lysosomal membrane protein Arl8, demonstrating that in living PGCs, DNase II is localized within lysosomes, while in the dying PGCs, it is released from the lysosomes.

The Top1-mediated ligation assay suggests that PGCs do exhibit dnase II-dependent DNA cleavage. Do the cells that exhibit dnase II cuts have a “condensed and distorted morphology”? Do cells with dnase II cuts co-stain for nuclear dnase II? The data in Fig 4 suggests that the frequency of Top1-mediated ligation events is lower in dnase II mutants. This is an important result and the results of the assays in Fig 4 should be quantified.

Although specific when it works, the robustness of the method for detection of DNase II-specific cuts (Top I-mediated ligation) is not optimal for quantitative detection in the dying PGCs, presumably due to the asynchronous nature of PGC death and the transient nature of this signal in the dying PGCs, which seems to appear after sufficient accumulation of DNA breaks and to disappear once the DNA is highly fragmented. Therefore, under these conditions, this method is rather qualitative than quantitative, and its strength in our system stems from the contrast between the positive labeling (albeit scarce) of the developmentally dying PGCs and the negative labeling of the apoptotic PGCs following Hid OE.

In contrast, staining for γ H2Av, which is an early marker of DNA breaks, was much more reproducible in our hands. Quantifications of the percentage of embryos with midline γ H2Av

positive PGCs in control and *dnaseII* mutants revealed a clear correlation between DNA breaks and functional DNase II (new Fig. 4b-d). Also, as proposed by the reviewer, we added new Supplementary Fig. 5d, depicting 2 dying PGCs found at mid and late demolition stages, clearly displaying localization of both DNase II and γ H2Av in the nucleus.

[3] The authors find that ATR/Chk1 mutants do not lose PGCs over time. How ATR/Chk1 contributes toward Dnase II dependent cell death is not clear. Cells that escape cell death in Chk1 mutants have nuclear Dnase II suggesting that ATR/Chk1 act downstream to nuclear translocation of Dnase II to initiate cell death. Do the cells with nuclear Dnase II in exhibit any Dnase II-cuts or show any evidence for DNA breaks? If so, this would strengthen the connection between nuclear dnase II and DNA cleavage. These cells should undergo apoptosis at a later stage. If not then the data would suggest that are necessary for the activation of endonuclease activity. This would be important.

We performed the suggested experiment of double staining *chk1* knockdown embryos for DNase II and γ H2Av. In contrast to control embryos where dying midline PGCs at mid-to-late demolition stages readily displayed both nuclear DNase II and γ H2Av, these events were highly rare (observed in one out of 50 embryos) in the *chk1* knockdown ectopically surviving midline PGCs, and which exhibited only partial nuclear translocation of DNase II (new Supplementary Fig. 5d,e). These findings suggest that the ATR/Chk1 branch is required for further translocation of DNase II to the nucleus in a feedback loop, but not for activation of the nuclease. Consistently, PAR release from the nucleus is attenuated but not blocked in the *chk1* knockdown embryos (Fig. 7g-i).

We note that despite displaying pronounced γ H2Av staining, the ectopically surviving midline PGCs in the *chk1* knockdown embryos displayed normal, non-apoptotic, morphology, implying that activation of the PGC death pathway might also deactivate the apoptotic pathway.

It is also noteworthy that as opposed to the partial nuclear translocation of DNase II in the *chk1* knockdown embryos, DNase II nuclear translocation was more pronounced and complete in the *bub3* OE ectopically surviving midline PGCs, suggesting that this checkpoint protein is further downstream to the ATR/Chk1 branch of the DDR, and might affect the execution stages of PGC death (Supplementary Fig. 7g,j).

We added in the *Discussion* section a paragraph speculating about how the DDR might act to promote and execute PGC death, and we also slightly updated the model (Fig. 8) to reflect the involvement of the ATR/Chk1 branch in the amplification loop.

[4] The requirement for AIF/PARP and the implication of parthanatos as the cell death program in PGCs.

The aif RI and the PARP RI mutant phenotypes with respect to preventing reduction of PGCs are not completely penetrant. Dnase II staining in aif/PARP mutants that show s that Dnase II fails to accumulate in the nucleus. Here again, quantitation of the frequencies of cells expressing nuclear Dnase II would be helpful to assess the contribution of AIF/PARP to the overall process.

In other words, is Dnase II accumulation observed in all mutants? It is possible that Dnase II translocation and activation in PGCs involves multiple genes including AIF/PARP.

As mentioned in our reply to point 2 of this reviewer, we now added quantifications of DNase II nuclear translocation in LMP mutants, and *aif* and *cyp1* knockdowns, showing that essentially almost all of the ectopically surviving midline PGCs in these mutants failed to accumulate nuclear DNase II, while in contrast, 70% of the condensed dying midline PGCs in control embryos displayed nuclear DNase II. The new data is presented in new Fig. 6h; new Supplementary Fig. 4j; new Supplementary Fig. 9e.

Although genes like AIF/PARP, with roles in parthanatos are required for the observed reduction in PGC number, whether the involvement of these genes shows that the program of cell death is parthanatos is not clear. It is plausible that the program in PGCs is a variant of this pathway, or a more complex program of cell death.

We agree that that AIF and PARP-1 alone might not be sufficient to define a cell death pathway as parthanatos. In particular that multiple studies implicated AIF in non-apoptotic cell death pathways without further investigating whether or not this is indeed a parthanatos pathway. However, in the current study, we dwelled deeper into the known mechanisms underlying parthanatos. We showed that AIF, PARP-1 and a nuclease (DNase II) are all involved in PGC death, as well as other components acting to facilitate their actions (e.g. LMP components, CypA, DDR pathway, etc). We presented evidence for both direct and genetic interactions between AIF and DNase II. We show that similar to parthanatos, the PGC death pathway is triggered by DNA damage rather than that the DNA damage merely participating in the final demolition stages, such as in apoptosis. We demonstrate that similar to parthanatos, which is mediated by AIF-promoting nuclear translocation of a nuclease (MIF), DNase II nuclear translocation requires functional AIF. Finally, we show that like in parthanatos, the release from the nucleus of the PARP-1 product, PAR, is directly correlated with PGC death. Altogether, we argue that PGC death is a subtype (perhaps the developmental subtype) of parthanatos.

In contrast, we showed that in addition to apoptosis, key components in other non-apoptotic developmental cell death pathways in *Drosophila*, such as the lysosomal-dependent cell death (LDCD)-like developmental pathway (called Germ Cell Death) and autophagy-dependent cell death (ADCD), are not involved in PGC death.

Arguably, the program of cell death in PGCs is an unusual example of non-apoptotic dnase II-dependent cell death with features of the parathanatos pathway. While the title of the paper “DNase II triggers a form of developmental cell death by parthanatos” may be reasonable, it may be an oversimplification.

We modified the title to more accurately reflect our findings as follows: “DNase II mediates a developmental subtype of parthanatos cell death in *Drosophila*”

[5] If any or all of the RNAi lines used in this study have been functionally validated in other studies then these studies should be cited.

We added citations to previously validated fly lines and also validated OE and knockdown of fly lines that were used for the first time in the current study (Fig. R1; a reviewer only figure at the end of this document).

[6] The titles of Figure 2 “Inactivation of the lysosomal endonuclease and compromised lysosomal biogenesis and leakage, all block PGC death” and Extended Figure 1 “PGC death is distinct from lysosomal-dependent or autophagy- dependent cell death” appear contradictory.

These titles are not contradictory but we understand why they might have been confusing. Germ Cell Death (GCD) is a Lysosomal-dependent cell death (LDCD)-like pathway in *Drosophila* (Yacobi-Sharon et al., 2013). We show that although GCD involves lysosomal and mitochondrial components, the key components in this pathway are not involved in PGC death by parthanatos. For clarity, we changed the title of Supplementary Fig. 1 to “PGC death is distinct from germ cell death (GCD) and autophagy-dependent cell death (ADCD).”

[7] Extended Figure 1j, please check numbers for nos>atg5,6,8 RI at ES13. The numbers reported and the graph do not tally.

Indeed, the total numbers of the examined PGCs for each fly line was mistakenly switched - it is now corrected (presented in Supplementary Fig. 1).

[8] The abbreviation OE requires to be defined.

It is now defined.

References

- Aits, S., and Jaattela, M. (2013). Lysosomal cell death at a glance. *J.Cell Sci.* *126*, 1905–1912.
- Benjamin, D., Robay, D., Hindupur, S.K., Pohlmann, J., Colombi, M., El-Shemerly, M.Y., Maira, S.M., Moroni, C., Lane, H.A., and Hall, M.N. (2018). Dual Inhibition of the Lactate Transporters MCT1 and MCT4 Is Synthetic Lethal with Metformin due to NAD⁺ Depletion in Cancer Cells. *Cell Rep.* *25*, 3047-3058.e4.
- Candé, C., Vahsen, N., Garrido, C., and Kroemer, G. (2004a). Apoptosis-inducing factor (AIF): caspase-independent after all. *Cell Death Differ.* *11*, 591–595.
- Candé, C., Vahsen, N., Kouranti, I., Schmitt, E., Daugas, E., Spahr, C., Luban, J., Kroemer, R.T., Giordanetto, F., Garrido, C., et al. (2004b). AIF and cyclophilin A cooperate in apoptosis-associated chromatinolysis. *Oncogene* *23*, 1514–1521.
- Deshpande, G., Calhoun, G., Yanowitz, J.L., and Schedl, P.D. (1999). Novel functions of nanos in downregulating mitosis and transcription during the development of the *Drosophila* germline. *Cell* *99*, 271–281.
- Feldstein, A.E., Werneburg, N.W., Li, Z., Bronk, S.F., and Gores, G.J. (2006). Bax inhibition protects against free fatty acid-induced lysosomal permeabilization. *Am. J. Physiol. - Gastrointest. Liver Physiol.* *290*.
- Hanyu-Nakamura, K., Kobayashi, S., and Nakamura, A. (2004). Germ cell-autonomous Wunen2 is required for germline development in *Drosophila* embryos. *Development* *131*, 4545–4553.
- Leach, T.J., Mazzeo, M., Chotkowski, H.L., Madigan, J.P., Wotring, M.G., and Glaser, R.L. (2000). Histone H2A.Z is widely but nonrandomly distributed in chromosomes of *Drosophila melanogaster*. *J. Biol. Chem.* *275*, 23267–23272.
- Renault, A.D., and Lehmann, R. (2006). Follow the fatty brick road: lipid signaling in cell migration. *Curr. Opin. Genet. Dev.* *16*, 348–354.
- Renault, A.D., Sigal, Y.J., Morris, A.J., and Lehmann, R. (2004). Soma-germ line competition for lipid phosphate uptake regulates germ cell migration and survival. *Science* (80-.). *305*, 1963–1966.
- Renault, A.D., Kunwar, P.S., and Lehmann, R. (2010). Lipid phosphate phosphatase activity regulates dispersal and bilateral sorting of embryonic germ cells in *Drosophila*. *Development* *137*, 1815–1823.
- Sano, H., Renault, A.D., and Lehmann, R. (2005). Control of lateral migration and germ cell elimination by the *Drosophila melanogaster* lipid phosphate phosphatases Wunen and Wunen 2. *J. Cell Biol.* *171*, 675–683.
- Slaidina, M., and Lehmann, R. (2017). Quantitative Differences in a Single Maternal Factor Determine Survival Probabilities among *Drosophila* Germ Cells. *Curr. Biol.* *27*, 291–297.
- Sonnenblick, B.P. (1941). Germ Cell Movements and Sex Differentiation of the Gonads in the *Drosophila* Embryo. *Proc.Natl.Acad.Sci.U.S.A* *27*, 484–489.
- Sonnenblick, B.P. (1950). The early embryology of *Drosophila melanogaster*. In *Biology of*

Drosophila, M. Demerec, ed. (New York: John Wiley), pp. 62–167.

Technau, G.M., and Campos-Ortega, J.A. (1986). Lineage analysis of transplanted individual cells in embryos of *Drosophila melanogaster*: III. Commitment and proliferative capabilities of pole cells and midgut progenitors. *Roux's Arch. Dev. Biol. Off. Organ EDBO* 195, 489–498.

Underwood, E.M., Caulton, J.H., Allis, C.D., and Mahowald, A.P. (1980). Developmental fate of pole cells in *Drosophila melanogaster*. *Dev. Biol.* 77, 303–314.

Wang, Y., An, R., Umanah, G.K., Park, H., Nambiar, K., Eacker, S.M., Kim, B., Bao, L., Harraz, M.M., Chang, C., et al. (2016). A nuclease that mediates cell death induced by DNA damage and poly(ADP-ribose) polymerase-1. *Science* (80-.). 354.

Yacobi-Sharon, K., Namdar, Y., and Arama, E. (2013). Alternative germ cell death pathway in *Drosophila* involves HtrA2/Omi, lysosomes, and a caspase-9 counterpart. *Dev. Cell* 25, 29–42.

Yamada, Y., Davis, K.D., and Coffman, C.R. (2008). Programmed cell death of primordial germ cells in *Drosophila* is regulated by p53 and the Outsiders monocarboxylate transporter. *Development* 135, 207–216.

Yuste, V.J., Moubarak, R.S., Delettre, C., Bras, M., Sancho, P., Robert, N., D'Alayer, J., and Susin, S.A. (2005). Cysteine protease inhibition prevents mitochondrial apoptosis-inducing factor (AIF) release. *Cell Death Differ.* 12, 1445–1448.

Fig. R1

Fig. R1 Validation of the knockdown and overexpression lines used in this study. a,b, Representative images of ES 10 WT (a) and PGC-specific *dnaseII* knockdown (b) embryos stained to visualize the PGCs (Vasa; red) and DNase II (anti-DNase II antibodies; green). The outlined areas (yellow squares) are magnified in the right panels. Scale bars, 50 μ m. (c-k) Quantitative RT-PCR analysis in embryos with maternal knockdown of *dor* (c), *atm* (d), *topbp1* (e), *orc2* (f), *cyp1* (g), *CG17266* (h), *CG2852* (i) and maternal overexpression of *orc2* (j) and *bub3* (k). (l) Western blot analysis of AIF protein levels in WT testes compared to testes with germ cell specific *aif* knockdown. Ponceau S stain was used as a loading control.

Fig. R2

Fig. R2 RNA expression patterns of the major mediators of PGC death in early embryonic stages. Images were adapted from two databases of *in situ* RNA hybridizations in *Drosophila* embryos; the BDGP *in situ* database (in blue) and Fly-FISH (DNA in red, RNA in green). Maternally deposited RNA (ES 1-3) and expression in early PGCs (ES 4-8) were detected for all the major mediators of PGC death.

REVIEWERS' COMMENTS

Reviewer #1 (Remarks to the Author):

The authors addressed all the questions raised in my review, I really appreciate the changes they made and strongly support publication.

Reviewer #2 (Remarks to the Author):

The authors have addressed the reviewer comments in a careful manner. In those cases where either new experiments, improved illustrations or revised statistical analyses were not possible, the authors have argued or explained in a satisfactory manner in their rebuttal why there were no changes made.

Overall this is an interesting paper. Most importantly, as stated in the final response to the reviewers, it firstly identify the physiological role of parthanatos in PGCs elimination during early embryogenesis.

Reviewer #3 (Remarks to the Author):

The revised manuscript is much clearer and is easier to follow. The data support the conclusions drawn with respect to the unconventional pathway of cell death in PGCs. In my view, the major criticisms have been satisfactorily addressed. I have some reservations about the use of the word "subtype" in the title as it could convey a degree of precision that is currently unwarranted but it is acceptable. The manuscript requires careful proof-reading. The TLA ACD features in the abstract but is not defined. There are typos in the abstract (line 27) and elsewhere.

Reviewer #1 (Remarks to the Author):

The authors addressed all the questions raised in my review, I really appreciate the changes they made and strongly support publication.

Thanks

Reviewer #2 (Remarks to the Author):

The authors have addressed the reviewer comments in a careful manner. In those cases where either new experiments, improved illustrations or revised statistical analyses were not possible, the authors have argued or explained in a satisfactory manner in their rebuttal why there were no changes made.

Overall this is an interesting paper. Most importantly, as stated in the final response to the reviewers, it firstly identify the physiological role of parthanatos in PGCs elimination during early embryogenesis.

Thanks

Reviewer #3 (Remarks to the Author):

The revised manuscript is much clearer and is easier to follow. The data support the conclusions drawn with respect to the unconventional pathway of cell death in PGCs.

Thanks

In my view, the major criticisms have been satisfactorily addressed. I have some reservations about the use of the word "subtype" in the title as it could convey a degree of precision that is currently unwarranted but it is acceptable.

The title was changed according to the Editor suggestion and the word "subtype" was removed.

The manuscript requires careful proof-reading.

Done

The TLA ACD features in the abstract but is not defined.

ACD is now defined in the abstract.

There are typos in the abstract (line 27) and elsewhere.

That typo was corrected (should have been "an" instead of "and"). We also carefully proof-read the manuscript.